# Comparison of surface-wave techniques to estimate S- and P-wave velocity models from active seismic data.

Farbod Khosro Anjom[1], Frank Adler[2], Laura Valentina Socco[1]

[1]Department of Environment, Land and Infrastructure Engineering, Politecnico di Torino, Turin, 10128, Italy
[2] CSTJF, TotalEnergies, Pau, 64000, France

*Correspondence to*: Farbod Khosro Anjom (Khosro-Anjom.Farbod@polito.it)

**Abstract.** The acquisition of seismic exploration data in remote locations presents several logistical and economic criticalities. The irregular distribution of sources and/or receivers facilitates seismic acquisition operations in these areas. A convenient approach is to deploy nodal receivers on a regular grid and to use sources only in accessible locations, creating an
irregular source-receiver layout. It is essential to evaluate, adapt, and verify processing workflows, specifically for near-surface velocity model estimation using surface-wave analysis, when working with these types of data sets. In this study, we applied three surface-wave techniques (i.e., wavelength/depth method, laterally constrained inversion, and surface-wave tomography) to a large-scale 3D data set obtained from a hard-rock site using the irregular source-receiver acquisition method. The methods were fine-tuned for the data obtained from hard-rock sites, which typically exhibit a low signal-to-
noise ratio. The wavelength-depth method is a data transformation method that is based on a relationship between skin depth and surface-wave wavelength and provides both S- and P-wave velocity ($V_s$ and $V_p$) models. We used Poisson's ratios estimated through the wavelength-depth method to constrain the laterally constrained inversion and surface-wave tomography and retrieve both $V_s$ and $V_p$ also from these methods. The pseudo 3-D $V_s$ and $V_p$ models were obtained down to 140 m depth over approximately 900 × 1500 m² area. The estimated models from the methods matched the geological
information available for the site. Less than 6% difference was observed between the estimated $V_s$ models from the three methods, whereas this value was 7.1% for the retrieved $V_p$ models. The methods were critically compared in terms of resolution and efficiency, which provides valuable insights into the potential of surface-wave analysis for estimating near-surface models at hard-rock sites.

## 1 Introduction

In order to overcome the difficulties of collecting seismic data in remote areas such as foothills and forests, a new acquisition method has been recently introduced, in which the nodal receivers are deployed in a regular grid, while the source locations are restricted to reachable areas such as the access roads (Lys et al., 2018). This approach creates an irregular source-receiver outline that raises the necessity to evaluate, verify, and test the seismic processing workflow. Here, we focus on the application of surface-wave methods to the data recorded through an irregular source-receiver scheme with the purpose of

near-surface velocity model estimations. These velocity models can be used for engineering purposes or as input in the exploration processing workflow to improve static corrections and groundroll removal.

Surface-wave methods are powerful tools for subsurface characterization. Most of these methods process the data to extract the surface-wave phase velocity dispersion curve (DC) from seismic records and invert these DCs individually to estimate the velocity models. Since the energy decay of surface-wave wavefields in depth depends on their wavelength, the investigation depth of surface-wave methods is related to the maximum recovered wavelength and can be considerably variable, ranging from a few meters (e.g., Xia et al., 2002; Feng et al., 2004; Comina et al., 2011; Pan et al., 2018) to several tens of meters (e.g., Mordret et al., 2014; Da col et al., 2019) or even to a few kilometers (e.g., Ritzwoller and Levshin, 1998; Kennet and Yoshizawa, 2002; Fang et al., 2015). The estimated models from surface-wave techniques can be used in many applications, such as near-surface site characterization (Lai, 1998; Xia, 2014; Foti et al., 2015), static corrections (Mari, 1984; Roy et al., 2010), and ground roll prediction and damping (Blonk and Herman, 1994; Ernst et al., 2002; Halliday et al., 2010).

Different methods can be adopted for extracting DCs from the seismic records and inverting them (see, for instance, Papadopoulou (2021) for a thorough review of the different processing techniques and characteristics of the estimated DCs). The retrieval of a velocity model from the DC can be based on simple data transformations or on model optimization approaches with different inversion strategies. According to the chosen workflow, the computational cost and model resolution may vary, and identifying the optimal approach for the analysis is an important task. Here, we compare three different methods (i.e., wavelength/depth data transform, laterally constrained inversion, and surface-wave tomography) that are rarely used for near-surface 3D model estimation. We apply these methods to a large-scale test data set acquired from irregular deployed source-receiver layout. The data were collected from a hard-rock site, which is typically characterized by a lower signal-to-noise ratio data compared to data from loose granular material (Papadopoulou, 2021).

Regardless of the type of surface-wave technique, since phase velocity DCs are known to have lower sensitivity to P-wave velocity ($V_p$) compared to S-wave velocity ($V_s$), most surface-wave methods focus on $V_s$ estimation, and they require *a priori* Poisson's ratio or $V_p$ for the inversion stage. Recently, a data transformation method based on the so-called wavelength-depth (W/D) relationship was developed to estimate both $V_s$ and $V_p$ models (Khosro Anjom et al., 2019). The W/D relationship is obtained by computing wavelength - depth couples corresponding to equal phase-velocity of surface-waves and time-average $V_s$ and represents the skin depth of surface-waves (Socco et al., 2017). Once estimated, the W/D relationship can be used to directly estimate the time-average $V_s$ from DCs. Socco and Comina (2017) showed with synthetic tests and tests on real data that the W/D relationship is highly sensitive to Poisson's ratio, and it can be used to estimate time-average $V_p$. Khosro Anjom et al. (2019) developed a data-driven W/D workflow that directly estimates interval $V_s$ and $V_p$ models from the DCs and is valid for sites with significant lateral variations. The method also provides Poisson's ratio, which can be used *a priori* in other surface-wave methods, such as laterally constrained inversion (LCI) and surface-wave tomography (SWT). It is important to note that there are substantial differences between layered, interval, and time-average velocities, which will be

frequently used in this paper. In a locally 1D velocity model, the interval velocity refers to the constant velocity of a layer between two specific depth levels. Here, we employ the term "interval velocity" to denote the velocity of the 1-meter intervals used for discretizing the models within the scheme of W/D method. The time-average velocity at a given depth $z$ is the weighted average velocity from the surface to the considered depth, for which the one-way time is equal to the one-way time of the layered velocity model to the same depth, and can be directly computed from the layered velocity model as:

$$V_{sz}(z) = \frac{\sum\limits_{n} h_i}{\sum\limits_{n} \frac{h_i}{V_{s,i}}}, \tag{1}$$

where $h$ and $n$ are the layers thickness and the number of layers down to the depth of z.

The earliest applications of LCI were on resistivity data (Auken and Christiensen, 2004; Wisén et al., 2005; Auken et al., 2005). The first successful application of the LCI to surface-waves was performed by Wisén and Christiansen (2005). Despite the LCI's capability as an effective tool for estimating near-surface models, its full potential in practical applications has not been fully exploited. In this technique, several multi-channel DCs available along a line or over an area are associated with local relevant 1D models and inverted simultaneously. The parameters of the 1D models are connected laterally and vertically through a set of constraints, whose strength controls the variations between model parameters at adjacent model points (Boiero and Socco 2010). As a result, consistent and smooth estimated pseudo 2-D or 3-D models are usually obtained from the LCI applications.

In the context of earthquake seismology, SWT is a well-established method for $V_s$ reconstruction of the crust and upper mantle (Wespestad et al., 2019; Bao et al., 2015; Boiero, 2009; Yao et al., 2008; Shapiro et al., 2005). Recently, a few authors showed the application of SWT for near-surface characterization using active (Da Col et al., 2019; Socco et al., 2014; Khosro Anjom et al., 2021) and passive data (Badal et al., 2013; Picozzi et al., 2009; Colombero et al., 2022).

In the literature, DC estimation methods are usually categorized into multi-channel and two-station methods, even though there are no theoretical or significant technical differences between the two approaches (Papadopoulou, 2021). The multi-channel technique is the most common approach, in which the recordings from an array of receivers (in a 2-D scheme) go through a wavefield transform (e.g., $f$-$k$, $f$-$v$, and $\tau$-$p$), and the DC is picked on the spectrum as the local maxima within the frequency band of surface-waves. The multi-channel processing stage is repeated to estimate DCs at different locations, which are then inverted individually (e.g., fast simulated annealing of Beaty et al., 2002) or simultaneously (e.g., LCI of Socco et al., 2009) to estimate the $V_s$ model. For SWT, DC are estimated using the two-station processing method, in which the receiver couples aligned with sources are considered to estimate many path-averaged DCs that are later inverted using a tomographic scheme to estimate directly (Boschi and Ekstrom, 2002; Fang et al., 2015; Boiero, 2009; Socco et al., 2014; Karimpour et al., 2022) or indirectly (Yoshizawa and Kennett, 2004; Shapiro and Ritzwoller, 2002; Yao et al., 2008) the $V_s$ model.

Here, we show the application of the three surface-wave methods (W/D, LCI, and SWT) to estimate both $V_s$ and $V_p$ models. The first two methods are based on multichannel DCs, whereas the latter relies on DCs from two-station method. We apply the methods to a challenging test data set that was acquired at a hard-rock site using the irregular source-receiver layout. The irregular source-receiver outline limits the use of conventional multi-channel processing methods. Conversely, the irregular combination of source-receiver is favorable for the estimation of many two-station DCs with different azimuthal angles, providing high data coverage and uniform azimuthal distribution of the paths, which is to SWT's advantage. The three surface-wave methods are customized for analyzing data with a low signal-to-noise ratio, which is often observed in collected data from hard-rock sites.

In this paper, we first introduce the site and describe the acquired data. Then, we explain the multi-channel and two-station DC estimation processing techniques. Then, we briefly describe the W/D, LCI, and SWT velocity model estimation methods and show their application to the data set. We use the W/D method to estimate the *a priori* Poisson's ratio required by LCI and SWT methods, which we then employ to transform their $V_s$ results into $V_p$. Finally, we compare the estimated models and the obtained resolution from the application of each method and compare the methods from an efficiency point of view.

## 2 Site description and field data set

The location is a limestone quarry in the province of Aurignac in the south of France (Fig. 1a). In Fig. 1b, we show the satellite view of the site superimposed with the elevation map of the area. From north-west to south-east, a significant natural (outside the pits) and human-made elevation (inside the pits) contrast is present, which can cause highly scattered surface-waves. In Fig. 1c, we show the geological map of the area from the website of French geological survey (BRGM). The central, eastern, and northern part of the site are characterized by stiff formations belonging to Thanetian and Sparnecian stages, primarily composed of stiff limestone and marl. In the western zone, recent loose deposits are present (Ypresian), creating a significant lateral variation between the east and west portions of the site. The very dense limestone with dolomite layers from Danian stage is outcropping in the north, outside of the investigated area, and is expected to be reached in shallow subsurface in the investigated zone.

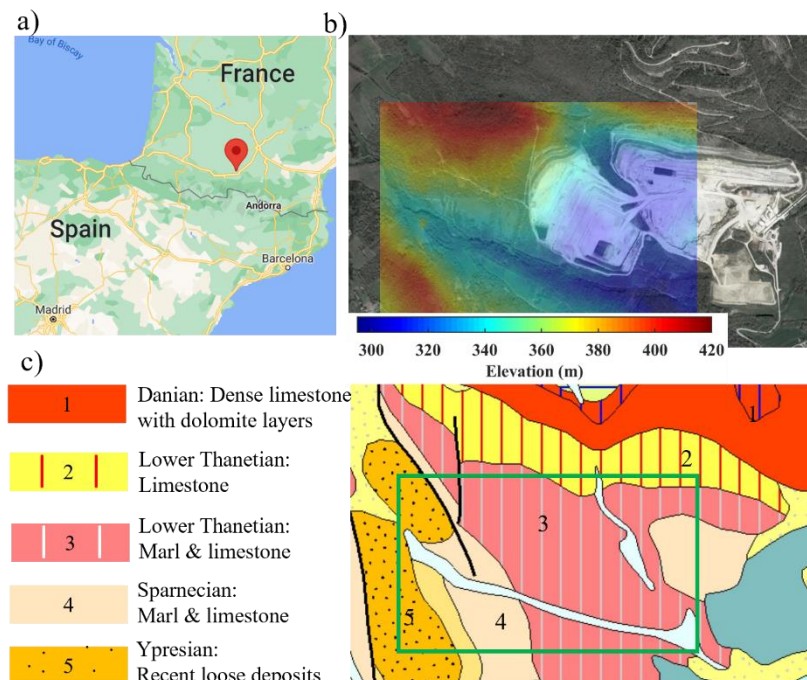

**Figure 1:** (a) Google map showing the location of the site (© Google Map). (b) Satellite view of the site obtained superimposed with the elevation map of the area (© Google Earth). (c) The geological map of the area obtained from French geological Survey (© BRGM - www.infoterre.brgm.fr). The green box shows the boundaries of the acquisition area.

The seismic campaign was conducted inside and outside the two open mining pits to test the irregular source-receiver layout acquisition technique at a hard rock site and provide an exploration data set to be used for testing different processing approaches. 918 receivers were deployed on a regular grid (area of 1.7 km × 1 km), and several source points (1077) were considered along the access roads, resulting in a 3-D large-scale acquisition layout. Two Birdwagen Mark IV off-road equipped with 24-ton vibrator were used as the source. The vibration included a sweep of 24 s (3 to 110 Hz) with 5 second

listening time. The data were collected in real-time using the RT2 wireless system. In this study, we considered a portion of the data that were collected outside the mining pits. The full description of the acquisition parameters corresponding to this portion of the data is given in Table 1.

**Table 1:** Acquisition parameters of the data set for outside the mining pits.

| Receivers | Sources | Number of receivers | Number of shots | Sampling rate (ms) | Recording time window (s) |
|---|---|---|---|---|---|
| 5 Hz vertical geophones | Vibroseis truck | 581 (spacing 25 and 50 m) | 533 (irregular layout) | 2 | 5 |

To minimize the effect of elevation contrasts (Fig. 1b), we split the data into two sub data sets (north and south), each corresponding to an area with relatively flat topography. In Fig. 2, we show the acquisition layout, where different colors are used for each sub data set.

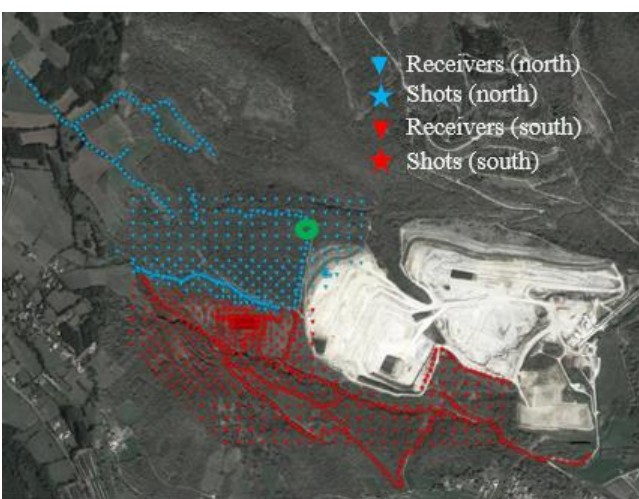

**Figure 2:** The satellite view of the Aurignac site (© Google Earth) superimposed with the acquisition layout. The data divided into two sub data sets shown with different colors, each within relatively flat area. The recordings from the highlighted shot (green circle) are plotted in Fig. 3.

In Fig. 3, we show the first 2 s of the recordings from the highlighted source in Fig. 2 in offset domain; only 20 % of the traces are shown for better visualization. The data exhibits a low signal-to-noise ratio as expected for hard rock sites.

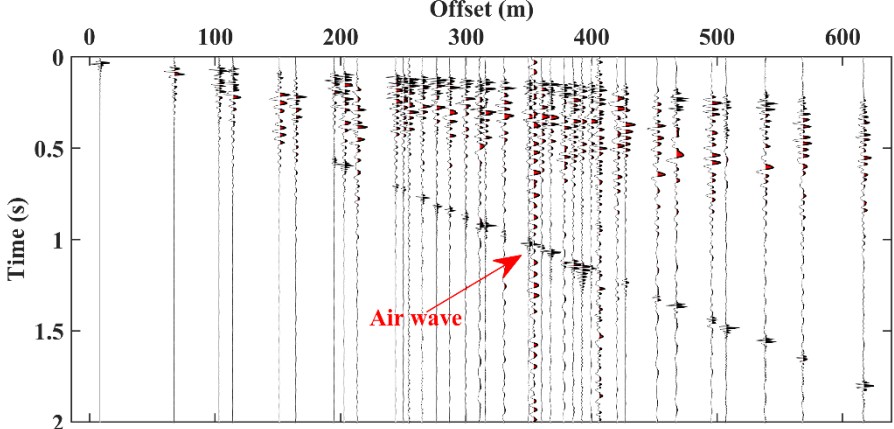

**Figure 3:** An example seismogram from the north zone of the Aurignac site. The shot location is highlighted with a green circle in Fig. 2.

## 3 Surface-wave processing of 3-D data

### 3.1 Multi-channel DCs

Multi-channel dispersion analysis is usually performed by selecting recordings from multiple inline receivers with a source and performing domain transform (i.e., $f - v, f - k, \tau - p$, etc.). Nevertheless, this approach can lead to inconsistencies in the estimated DCs when the technique is applied to 3-D data sets with irregular source-receiver geometry at sites with significant lateral variations. In Fig. 4, we show two examples of DC estimation from the field data set for the same location using recordings from two linear receiver arrays. The estimated DCs for the location shown as green × in Fig. 4b are on average more than 15% different. The main reason for this inconsistency is the impact of lateral variations and the entirely different surface-wave propagation path along the two arrays.

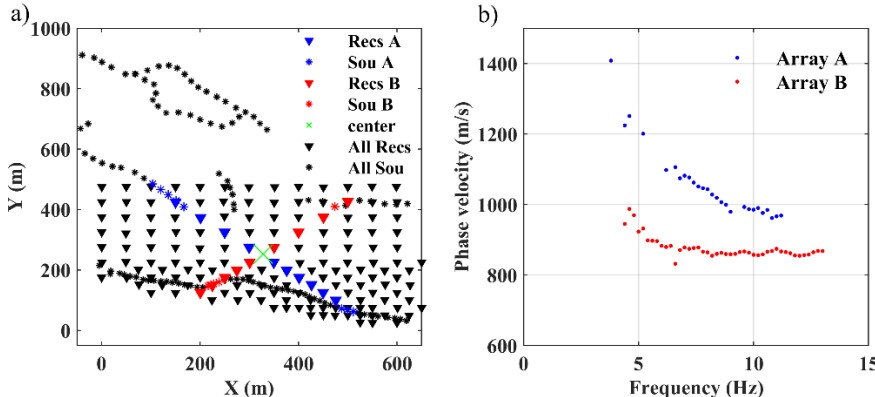

**Figure 4:** Two example DC estimations for the same location using two different receiver arrays. (a) The two receiver arrays and aligned shots. The middle of both arrays is coincident and shown with the green ×. (b) The estimated DCs for the location of the green × in a using the two arrays' records.

To minimize the impact of lateral variability on the DC estimation of the 3-D data, we consider the recordings from receiver
spread over an area (Wang et al., 2015; Xia et al., 2009; Park, 2019). For each DC estimation, we select the receivers inside a
square area (window) and consider the sources within a certain distance from the center of the square. We use the phase-shift
method (Park et al., 1998) to estimate the *f-v* spectrum. We stack the spectra corresponding to the recordings of the same
receivers but from different source locations to increase the signal-to-noise ratio (Neducza, 2007). Then, the fundamental
mode is picked as the maxima on the spectrum and is assigned to the center of the receiver spread. We slide the window by
170 one inter receiver spacing to estimate DCs corresponding to different locations of the site.

For the field data, we considered a window of $100 \times 100$ m$^2$ to select the receiver and sources within 250 m of the center of
the square. In Fig. 5, we show an example DC picking for the north zone. In Fig. 5a, the 9 selected receivers inside the
square are shown in blue, and the selected shots are plotted in green asterisks. In Fig. 5b, we show the computed spectrum
and the picked DC. Overall, we estimated 545 DCs for northern and southern zones shown in Fig. 5c.

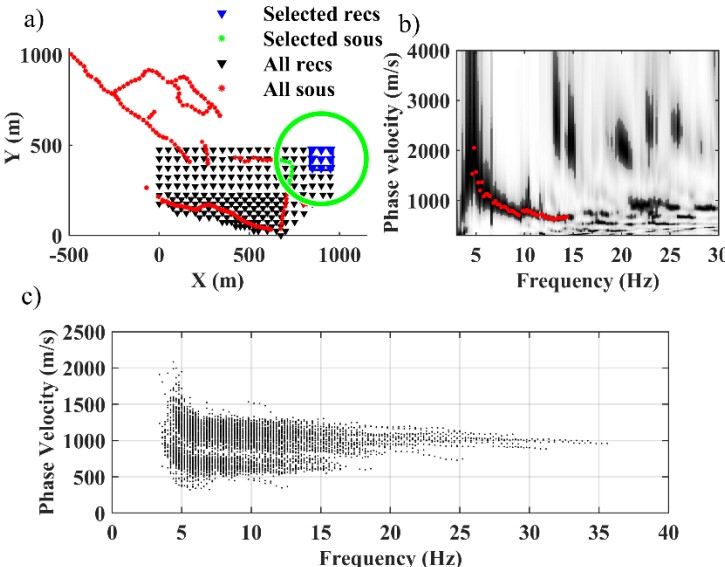

**Figure 5:** Multi-channel DC estimation from the field data. (a) An example geometry of the selected receivers and sources for DC
estimation in the northern zone. (b) The computed spectrum and estimated DC from the recordings of the receivers shown in (a). (c) All
the estimated DCs for the north and south zones.

## 3.2 Two-station DCs

The two-station DCs are estimated applying interferometry to the recordings from receiver couples aligned with a source,
assuming straight ray approximation. We use the algorithm developed by Da Col (2019) and modified by Khosro Anjom et
al. (2021). First, an automatic search is performed to find the receiver couples aligned with the source at each azimuth angle,

considering 1° tolerance for the deviation from a straight path. Given the scale of the site, we consider the propagation path as occurring over a plain area, and we neglect the great-circle approximation. Then, the traces are narrow band filtered at various frequencies, using zero-phase Gaussian filters and the filtered traces of the receiver couples are cross-correlated and assembled to form the cross-multiplication matrix. We use a 3$^{rd}$ order spline interpolator to convert the cross-multiplication matrix to the frequency-velocity domain. We stack the cross-multiplication matrices computed from the records of the same two stations but different sources, to increase the signal-to-noise ratio. Finally, at each frequency, the phase-velocity is picked as the maximum of the cross-multiplication matrix. To avoid cycle skipping, we use, as reference, the closest multichannel DC, and we automatically pick maxima closest to the reference DCs. To minimize the contamination of the fundamental mode by higher surface-wave modes, we damp the higher mode data using the muting strategy of Khosro Anjom et al. (2019).

We applied the two-station DC estimation method to the data from the north of the site only (blue markers in Fig. 2). We performed an automatic search of the receiver couples aligned with sources within 250 m offset, which resulted in 4710 possible receiver couples and source settings. We used the local DCs from the multi-channel analysis (Fig. 5c) as references to locate the correct trend of the path-averaged fundamental mode DCs. We discarded noisy or inconsistent cross-multiplication matrices and in total, 1301 path-averaged DCs were estimated. In Fig. 6a and b, we show the estimated path-averaged DCs and the observed azimuthal illumination. The data show uniform coverage with most paths showing angles between 0° to 40° and 140° to 180° angles. The uniform coverage mitigates the directionality of the tomographic inversion toward dominant directions (Khosro Anjom et al., 2021).

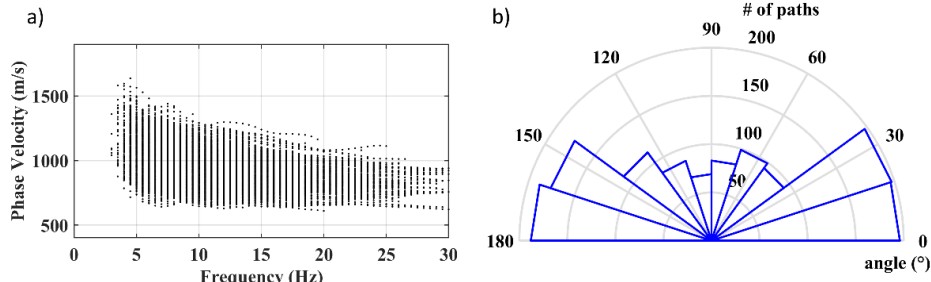

**Figure 6:** (a) The estimated path-averaged DCs corresponding to northern part of the site. (b) The obtained azimuthal illumination with the numbers around the great circle showing the angles and other circles representing the obtained coverage.

In Fig. 7a to f, we show the data coverage within different wavelength ranges, where the color scale shows the path-average phase-velocity. The data exhibit very high coverage for wavelengths between 40 to 220 m, beyond which it decreases substantially.

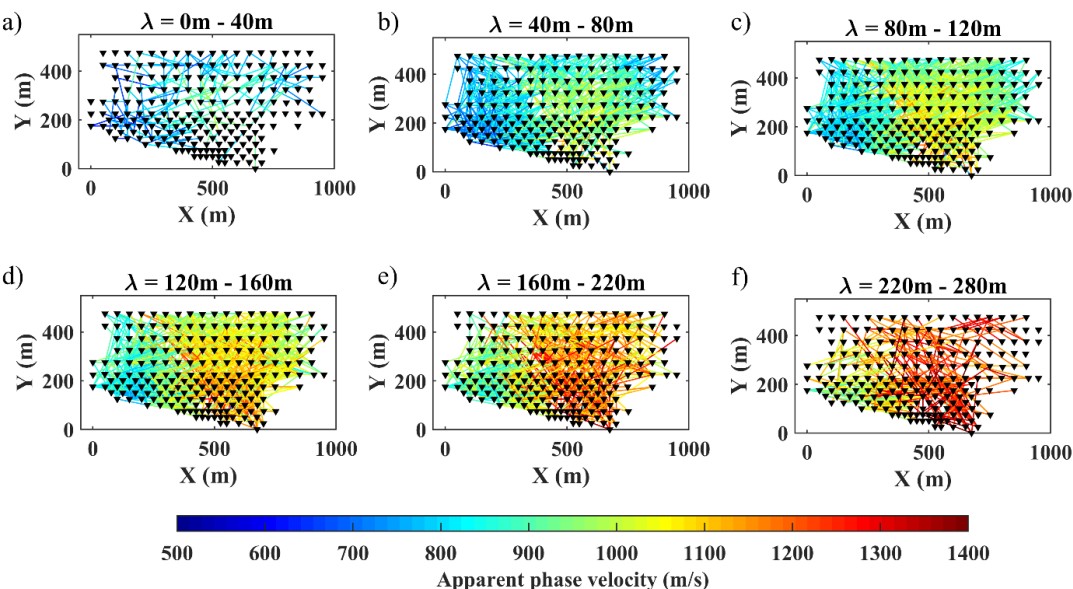

**Figure 7:** Pseudo-slices of the estimated path-averaged DCs from the north of the site shown within the wavelength ranges: (a) 0 to 40 m. (b) 40 to 80 m. (c) 80 to 120 m. (d) 120 to 160 m. (e) 160 to 220 m. (f) 220 to 280 m.

## 4 Velocity model estimation

### 4.1 W/D data transform

The only inputs of the W/D method are the estimated multi-channel DCs. The method, as described in Khosro Anjom et al. (2021), is composed of four main steps: (i) The clustering of DCs, (ii) the selection of a reference DC for each cluster and the estimation of the corresponding time-average $V_s$ velocity model, (iii) the W/D relationship and apparent Poisson's ratio estimation for the reference DC of each cluster, and (iv) the direct transformation of the DCs into time-average and then interval $V_s$ and $V_p$ models.

Since the same W/D relationship is applied to different DCs to transform them into velocity models, to apply the method at sites with significant lateral variations, the DCs must be clustered into more homogenous sets, and one W/D relationship should be estimated and applied separately to each cluster of DCs. We use the hierarchical clustering algorithm developed by Khosro Anjom et al (2017) to cluster the DCs.

For each cluster, a reference DC and its corresponding time-average $V_s$ model is needed to estimate the W/D relationship. The reference DC is selected based on the quality control proposed by Karimpour (2018). The time-average velocity is used in many applications, ranging from static correction of the reflection data to seismic hazard analysis where the time average velocity down to 30m of depth, the so called $V_{s30}$ , is used as a proxy for seismic response classification. To estimate the required time-average $V_s$ model, we invert the reference DC using an optimized Monte Carlo inversion (Socco and Boiero,

2008). The density has a minor impact on the surface-waves velocity (Xia et al., 1999; Foti and Strobbia, 2002) and is defined *a priori* based on the geological information about the site. The $V_s$, the Poisson's ratio, and the thicknesses are randomly sampled within the boundaries of a wide uniform model space. Then, the synthetic DCs corresponding to each

model are computed and compared with the experimental DC. The algorithm uses the scaling properties of DCs to create an optimal model space based on the initial model space: before computing the misfit with the experimental DC, the synthetic DCs of the random models are shifted as close as possible to the experimental DC; then, the scaling factor is obtained from the DC shift and is used to scale the models. These scaling steps, which are performed in a fully automatic manner, highly optimize the model space sampling that is focused on low misfit regions and reduce the number of required simulations.

Unlike deterministic inversions that result in a single output, the Monte Carlo inversion leads to a set of possible solutions. The best fitting models are selected according to a one-tailed Fisher test, imposing a certain level of confidence. Then, the selected layered $V_s$ models are transformed to time-average $V_s$ models using Eq. 1. The values of the selected time-average $V_s$ models are averaged at each depth to obtain a unique time-average $V_s$ model corresponding to the reference DC. Next, we estimate the W/D relationship that consists of the pairs of wavelength and depth values for which the phase velocity of the

DC and the time-average $V_s$ of the Monte Carlo solution have the same value. Comprehensive synthetic and real data analyses have been performed by Socco et al. (2017), Khosro Anjom et al. (2019), and Khosro Anjom (2021) to show the high sensitivity of the W/D relationship to Poisson's ratio. We use the method of Socco and Comina (2017) to estimate from the experimental W/D relationship an apparent Poisson's ratio that relates the time-average $V_s$ and time-average $V_p$: First, we generate synthetic DCs corresponding to the estimated $V_s$ model from the Monte Carlo inversion and different Poisson's

ratios. Then, we consider these DCs and the time-average $V_s$ to retrieve synthetic W/D relationships that are each corresponding to a specific Poisson's ratio value. Next, we deduce an apparent Poisson's ratio at each depth by comparing the experimental W/D relationship with the synthetic ones. The estimated Poisson's ratio is an apparent one that relates time-average $V_s$ and $V_p$ of the cluster (Khosro Anjom et al., 2019).

We use the estimated experimental W/D relationship to directly transform all DCs of the cluster into time-average $V_s$

models. Then, we transform the estimated time-average $V_s$ models into time-average $V_p$ through the estimated apparent Poisson's ratio. The time-average velocities can be transformed to interval velocities using a Dix-type equation. In order to reduce the impact of noise in the data, we employ the regularized Dix-type formulation suggested by Khosro Anjom (2019) to convert the time-average $V_s$ and $V_p$ models into interval $V_s$ and $V_p$. It is important to emphasize that the estimated models are not layered velocity models, but rather interval velocity models with 1-meter intervals for the entire investigation depth.

Finally, we assemble all the estimated models from the clusters to build a pseudo 2-D/3-D model.

**4.2 LCI**

The method's inputs are the multi-channel DCs and the initial models at the location of the local DCs. An initial model defined as the thickness, density, Poisson's ratio, and $V_s$, is set for each cluster of DCs. The thickness and $V_s$ are based on the

Monte Carlo inversions of reference multi-channel DCs in section 4.1. The Poisson's ratio is selected based on the W/D

analysis. We use information from the site to define the densities of the model.

The inversion method is a deterministic least-square inversion based on Auken and Christiansen (2004), which was developed by Boiero (2009) and modified by Khosro Anjom (2021) to support parallel computing. At each iteration the $V_s$ and thicknesses are updated, and the Poisson's ratio and density are fixed *a priori*. All the DCs are inverted simultaneously for a set of 1D models that are tied by lateral constraints between parameters of neighboring models. We use damped least

square inversion scheme (Marquart, 1963) with lateral constraints (Auken and Christiansen, 2004) to update the model iteratively. The constraints act as spatial regularization, and their strength is defined to avoid both overfitting and over smoothing. Week lateral constraints or lack of lateral constraints can create unrealistic lateral changes in the final model, whereas too strong constraints can result in an oversmoothed model, masking the sharp lateral variations in the site. We use the data misfit as an indicator for choosing the level of constraints: the inversion with the highest level of constraint that does

not impact the DC misfit compared to unconstrained inversion is selected. A thorough description of the method is available in Boiero (2009) and Socco et al. (2009) and the strategy for constraints selection is provided in Boiero and Socco (2010).

### 4.3 SWT

The inputs of the tomographic inversion are the path-averaged DCs from two-station method and the initial model. The

parameters of the initial model are the thickness, $V_s$, Poisson's ratio, and density. The model points are defined with equal distances in X and Y directions. The distance of the model points depends on the required resolution and also the data coverage (i.e., path-averaged DCs). The parameters of the initial model are selected the same as the initial model of the LCI (section 4.2).

We use the tomographic inversion algorithm developed by Boiero (2009) and modified by Khosro Anjom et al. (2021). An

essential part of the tomographic inversion is the computation of synthetic path-averaged DCs corresponding to the observed ones. We compute the path-averaged DCs, assuming a straight ray path approximation between the two receivers, and as reciprocal of the average slowness along the paths discretized over the model grid. The phase velocities at the location of the discretized paths are computed by bi-linear interpolation of the phase velocities from local DCs corresponding to the adjacent model points (Boiero, 2009).

Similar to the LCI algorithm, a damped least-least square method (Marquart, 1963) with lateral constraints is used to iteratively update the model until the minimum misfit between synthetic and observed DCs is reached. The only parameter that updates in the inversion is $V_s$, while the others are fixed *a priori*. The method allows the implementation of lateral constraints. We consider the same criteria explained in section 4.2 to select the optimal constraint level. In contrast with the other two methods, the SWT method is applied to the northern data set only due to computational capacity restrictions that

will be explained in the discussion in section 7.

## 5. Results

### 5.1. W/D data transform

The clustering of all the estimated DCs generated two clusters. In Fig. 8a, we show the estimated DCs with the color scale based on the clustering of the DCs in Fig. 8b. The DCs of the western cluster (cluster A, shown in blue in Fig. 8a) present lower phase-velocities compared to the eastern DCs (cluster B, shown in green in Fig. 8a).

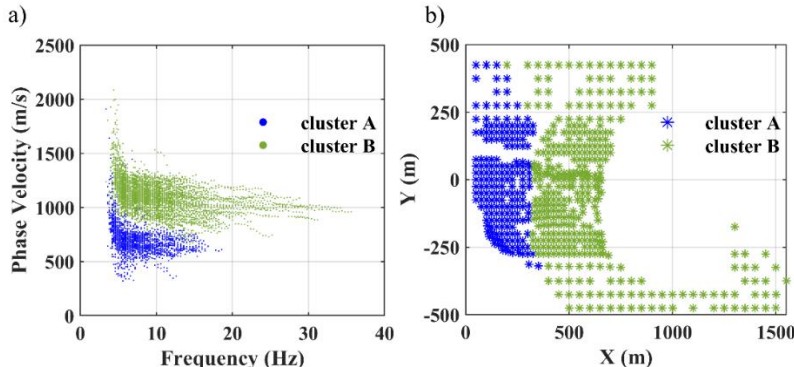

**Figure 8:** Clustering of the multi-channel DCs. (a) The estimated DCs. (b) The spatial view of the estimated DCs and obtained clusters.

In Fig. 9 and 10, we show the steps of estimating the reference W/D relationship and apparent Poisson's ratio for the reference DCs of cluster A and B. We considered variable Poisson's ratios between 0.1 and 0.45 for the Monte Carlo inversion. Based on the information from the site, we considered density of 2000 kg/m$^3$ for the first layer and constant density of 2200 kg/m$^3$ for the other layers. We imposed a 0.001 level of confidence for the Fisher test to accept the best-fitting models among 1,000,000 sampled models. In Fig. 9b and c, as well as Fig. 10b and c, we show the estimated $V_s$ and time-average $V_s$ models for cluster A and B, respectively. In Fig. 9b and Fig. 10b, we also show the boundaries of the $V_s$ model space for the Monte Carlo inversion. Due to the application of the scale properties, the selected models can be scaled to outside of the original boundaries of the model space (Socco and Boiero, 2008). The estimated W/D relationship for cluster A and B are shown in Fig. 9d and 10d, whereas the obtained apparent Poisson's ratios are provided in Fig. 9e and 10e. In Fig. 9d and e, as well as Fig. 10d and e, we also show the uncertainty associated to the reference W/D and to the apparent Poisson's ratio of the clusters, which was obtained based on the method of Khosro Anjom et al. (2019).

For both clusters, the W/D relationship and apparent Poisson's ratio were not available for the first 20 m due to the lack of short wavelength data in the experimental DCs. The investigation depth of 128 m was reached for cluster A, whereas this value was increased to 140 m for cluster B.

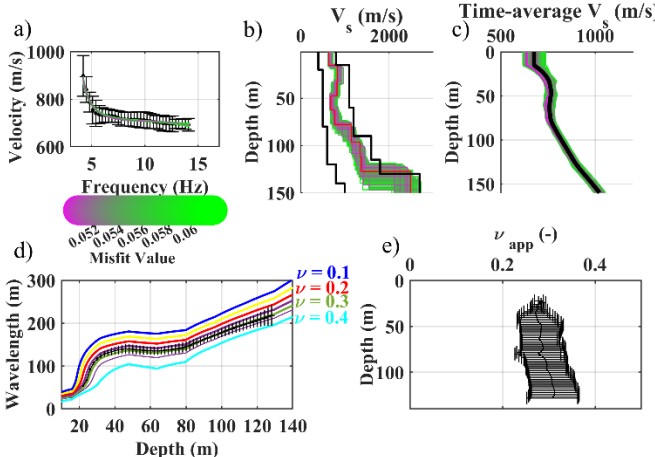

**Figure 9:** The steps of estimating the reference W/D relationship and apparent Poisson's ratio for cluster A. (a) The reference DC with the phase-velocity and uncertainty and the synthetic accepted DCs from the Monte Carlo inversion. (b) The accepted $V_s$ models from the Monte Carlo inversion, where the black lines show the initial boundaries of the model space. (c) The accepted time-average $V_s$ models from the Monte Carlo inversion. In black, the reference time-average $V_s$. (d) Estimated reference W/D relationship. The colored W/D relationships are the synthetic ones, each with constant Poisson's ratio, used for apparent Poisson's ratio estimation. (e) The estimated reference apparent Poisson's ratio.

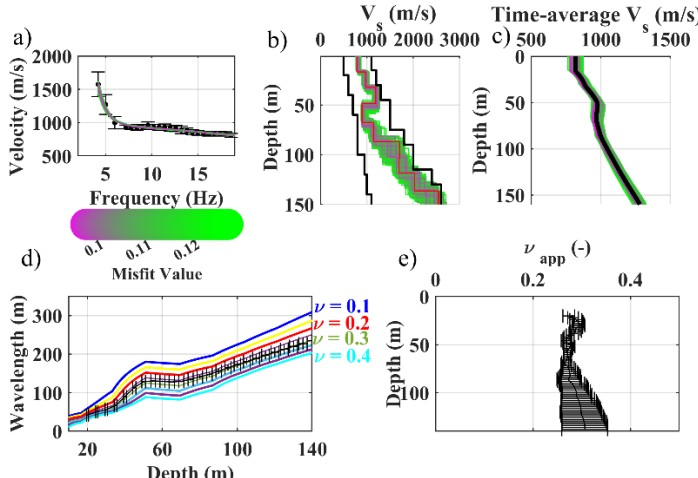

**Figure 10:** The steps of estimating the reference W/D relationship and apparent Poisson's ratio for cluster B. (a) The reference DC with the phase-velocity and uncertainty and the synthetic accepted DCs from the Monte Carlo inversion. (b) The accepted $V_s$ models from the Monte Carlo inversion, where the black lines show the initial boundaries of the model space. (c) The accepted time-average $V_s$ models from the Monte Carlo inversion. In black, the reference time-average $V_s$. (d) Estimated reference W/D relationship. The colored W/D relationships are the synthetic ones, each with constant Poisson's ratio, used for apparent Poisson's ratio estimation. (e) The estimated reference apparent Poisson's ratio.

The estimated DCs from the two clusters were transformed to interval $V_s$ and $V_p$ models using the reference W/D relationships and apparent Poisson's ratios (Fig 9d and e, and Fig 10d and e). In Fig. 11, we show several horizontal slices of the estimated $V_s$ model averaged over the depth intervals indicated on top of each plot. Similarly, in Fig. 12, we show the horizontal slices of the estimated $V_p$ averaged over the same depth intervals. Both models show a sharp velocity transition

between the east and west side of the area in shallow depths above 65 m (Fig. 11a, b and c and Fig. 12a, b and c). This contrast is created by the transition from the high-velocity limestone and marl formations in the east to loose materials in the west characterized by lower velocity. Below, 110 m (Fig. 11e and f, and Fig. 12e and f) the contrast disappears, reaching the high-velocity formation probably from the Danian stage.

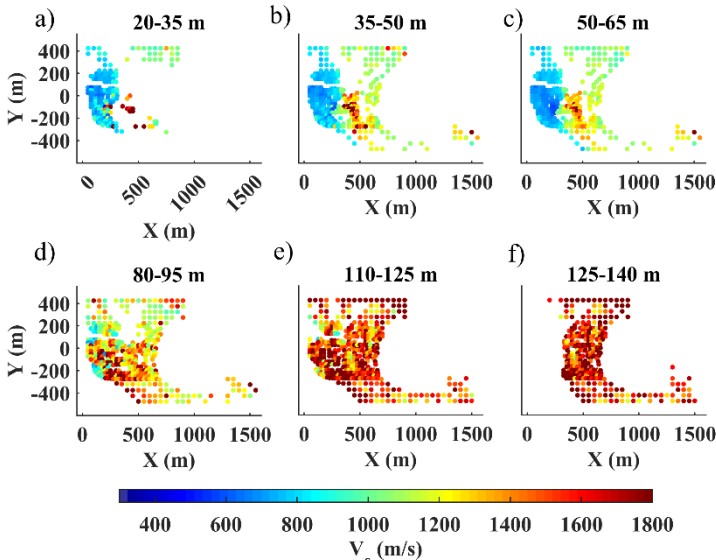

**Figure 11:** The estimated $V_s$ model using the W/D method. (a to f) The horizontal slices at different depth intervals indicated on top of each plot.

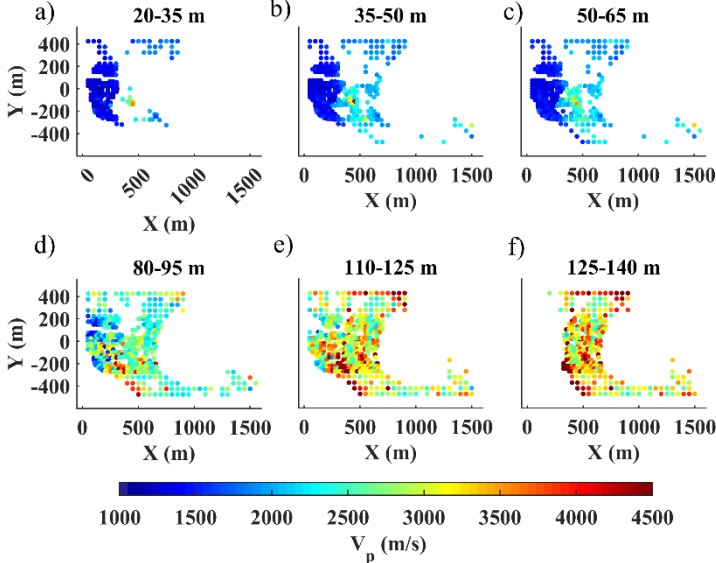

**Figure 12:** The estimated $V_p$ model using the W/D method. (a to f) The horizontal slices at different depth intervals indicated on top of each plot.

## 5.2. LCI

We defined an initial model composed of 9 layers overlying half-space with constant thicknesses of 15 m, except for the first layer which was set at 20 m, giving an investigation depth of about 140 m. We assigned the Poisson's ratios of the model based on the results of the W/D analysis: using the estimated $V_s$ and $V_p$ models at the reference location of each cluster (Fig. 13a and b), we obtained the Poisson's ratios, shown in Fig. 13c and d (in blue), corresponding to cluster A and cluster B, respectively. Since the Poisson's ratios are assumed invariant within each cluster, we used all estimated Poisson's ratios of each cluster from the W/D method (Fig. 13e) to obtain an uncertainty for the estimated Poisson's ratios at each depth. The horizontal error bars in Fig. 13c and d show the standard deviation of the estimated Poisson's ratios. We averaged and extrapolated the values of the Poisson's ratio to match them with the layers of the LCI model (red lines in Fig. 13c and d). Based on the clustering analysis of the W/D method (Fig. 8b) and location of the LCI model points, we assigned the appropriate Poisson's ratio to each 1D model. We defined a constant density of 2200 kg/m$^3$, except for the first layer (2000 kg/m$^3$) based on the geological formations in the area.

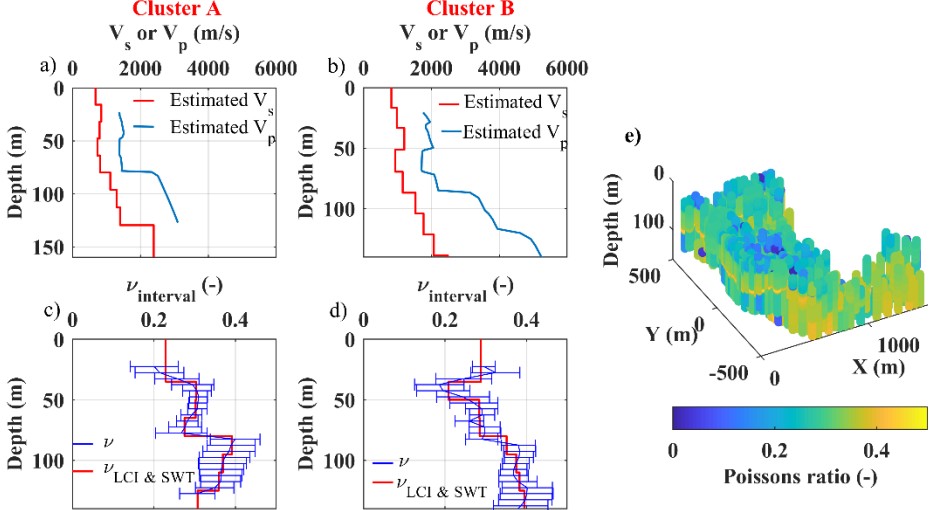

**Figure 13:** Poisson's ratio estimation for the cluster A and B. (a and b) The estimated $V_s$ and $V_p$ models corresponding to the reference DC of Clusters A and B. (c and d) In blue, the estimated Poisson's ratio for the cluster A and B. In red, the averaged and extrapolated Poisson's ratio corresponding to the layers of LCI and SWT. (e) The 3D view of the obtained Poisson's ratios from the estimates $V_s$ and $V_p$ models of the W/D method in Figures 11 and 12.

We performed an unconstrained and several laterally constrained inversions to find the optimal level of constraints according to the strategy described in Boiero and Socco (2010). We chose a lateral constraint on $V_s$ equal to 50 m/s, that was the highest level of constraints that did not significantly impact the inversion's residual misfit. The unconstrained inversion yielded a least square weighted misfit of 23.4, whereas the selected constrained inversion resulted in a misfit of 23.9. In Fig.

14, we show the horizontal slices of the estimated $V_s$ model at various depths. Even though the inversion is laterally constrained, the algorithm was able to depict a sharp transition between the east and west (Fig. 14a to c), which is in line with the results of the W/D method (Fig 11a to c). The LCI model below 87.5 m shows high velocities with insignificant variations between east and west.

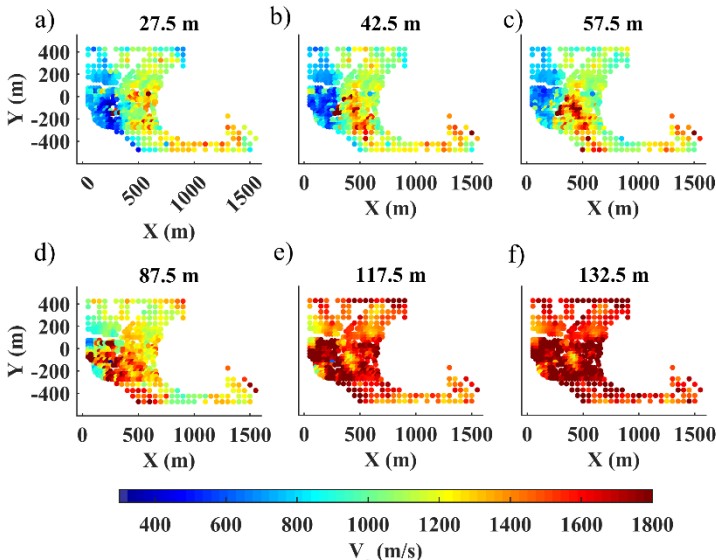

**Figure 14:** The estimated $V_s$ model using the LCI method. (a to f) The horizontal slices at different depths indicated on top of each figure.

### 5.3 SWT

Given the high data coverage from the estimated path-averaged DCs (Fig. 7), we defined a dense model grid on the considered northern zone, composed of 300 1D models, aiming at obtaining high resolution model. We used the same initial models defined for the LCI (section 5.2).

In Fig. 15, we show the estimated $V_s$ at the different layers, which is in correspondence of the depth intervals defined for the estimated $V_s$ of W/D method (Fig. 11). We chose a lateral constraint of $V_s$ equal to 50 m/s, that was the highest level of constraints that did not significantly impact the inversion's residual misfit. The unconstrained inversion yielded a least square weighted misfit of 42.1, whereas the selected constrained inversion resulted in a misfit of 43. Similar to the estimated model from W/D and LCI, the estimate model from SWT shows a significant velocity contrast between the east and west. Nevertheless, this contrast is smoother than the other two models. The model shows high velocities with no significant lateral variation below 110 m of depth, an indication of reaching the high-velocity limestone/marl formation from the Danian stage (Fig. 1c).

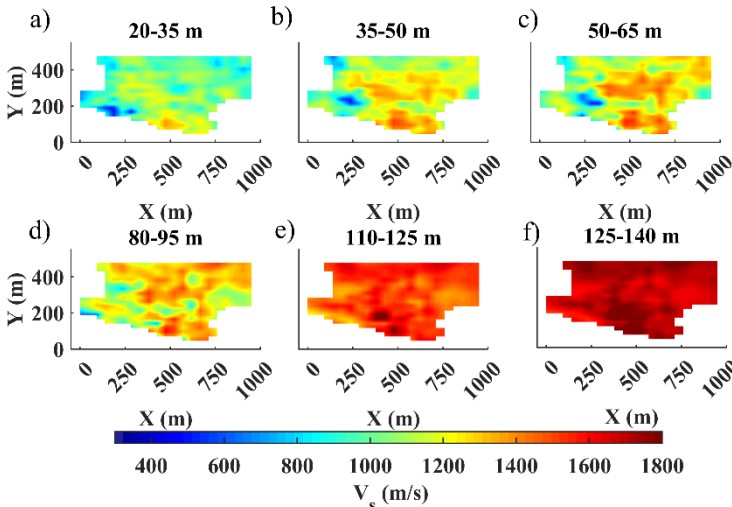

**Figure 15:** The estimated $V_s$ model for the north of the site using the SWT method. (a to f) The horizontal slices at different layers indicated on top of each figure.

## 7 Discussion

We showed the application of three surface-wave methods for $V_s$ model estimation, out of which, the W/D and LCI methods provided the velocity models at the location of DCs, both in the northern zone (174 locations) and southern zone (371 locations). The W/D method provided the $V_p$ model and the reference Poisson's ratios used in LCI and SWT. The SWT was applied only to the northern zone, which provided the $V_s$ models at 300 defined model points. Here, we further evaluate the results of the three methods in terms of vertical resolution, spatial resolution, differences of the estimated models and computational efficiency of each method.

In Fig. 16, we show the wavelength distribution of the estimated multi-channel DCs in blue that shows dense data sampling up to 300 m of wavelength, suggesting good vertical resolution also in deeper portions. These DCs were obtained from the recordings of receivers spread over a square window of $100 \times 100$ m$^2$. The receiver window was shifted by one receiver spacing (50 m in the north and 25 m in the south). Neglecting the smoothing effect of superimposed receiver windows, the shifting distance can be considered as the spatial resolution of the multi-channel DC data.

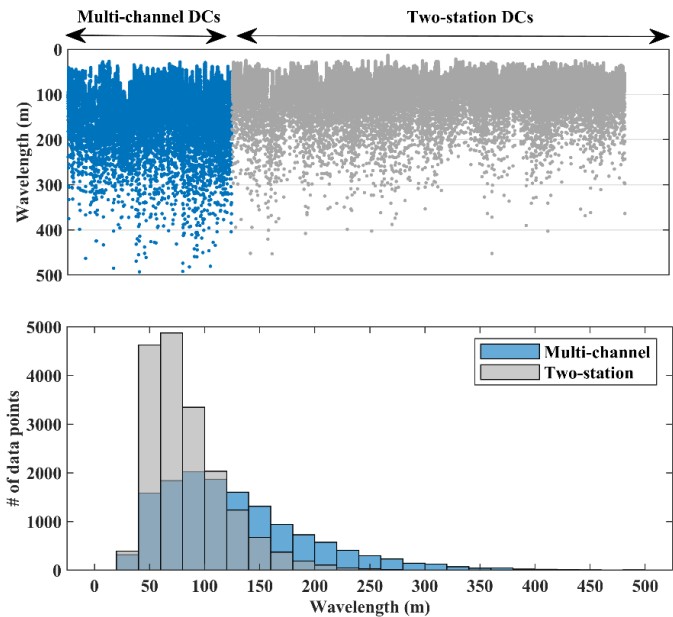

**Figure 16:** Comparison between the wavelength distributions of the multichannel and two-station dispersion data. (Top) The distribution of the wavelength shown separately for each estimated DC. (Bottom) The histogram showing the wavelength distribution of all DCs within 20 m wavelength bins.

In Fig. 16, in grey, we also show the wavelength distribution of the estimated two-station DCs. Even though the total number

of DCs from two-station analysis (1301) is far more than multi-channel ones (545), the large wavelength datapoints (>120 m) are sparser than the ones obtained with the multi-channel method (top plot in Fig. 16), mainly due to low signal-to-noise ratio of the cross-multiplication matrices at low frequencies. This shows that multi-channel DC analysis provides greater investigation depth compared to the two-station method. To mitigate the lack of investigation depth of SWT, we employed the wavelength-based weighting developed by Khosro Anjom et al. (2021), to increase the score of large-wavelength data

points in the tomographic inversion, aiming at enhancing the resolution at depth. We performed a checkerboard test to evaluate horizontal and vertical resolution of the SWT. We perturbed the estimated $V_s$ model from SWT method (Fig. 15) by 8% negatively and positively which alternated every two layers (Fig. 17a and b). In Fig. 17c to f, we show the results of the inversion at various layers. The inversion is effective in all considered depths and across the whole model. The $50 \times 50$ m$^2$ perturbations were well-recovered up to depth of 50 m (Fig. 17c and d), providing similar spatial resolution compared to LCI

and W/D methods. The resolution slightly decreases towards the deepest portion of the model (Fig. 17e and f), especially in the northern part where long-wavelength data are lacking (Fig. 13f). It is noteworthy to mention that the location of the circles in Fig. 17 matches the model grid used for the tomographic inversion of the real data (Figure 15).

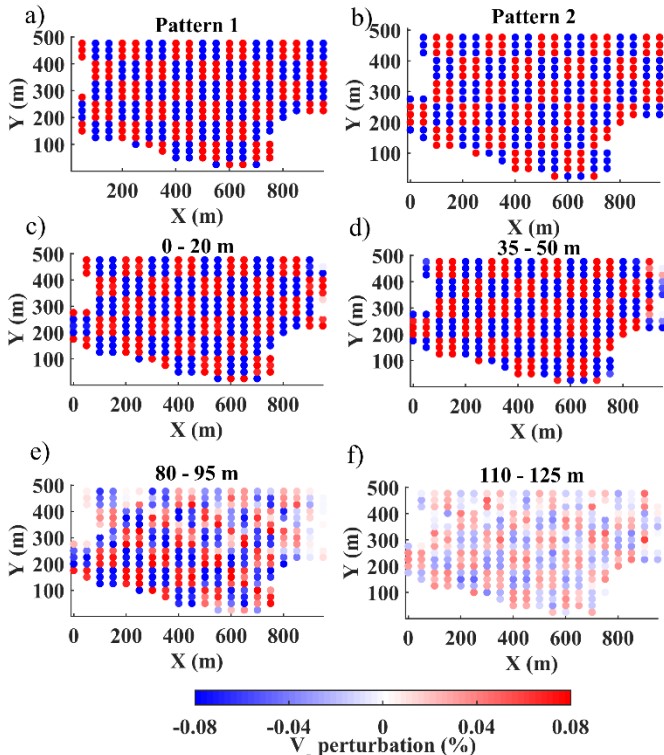

**Figure 17:** Checkerboard test. (a) The pattern 1 used to perturb $V_s$ of layer 1, 2, 5, 6, and 9. (b) The pattern used to perturb $V_s$ of layer 3, 4, 7, and 8. (c) 0 to 20 m. (d) 35 to 50 m. (e) 80 to 95 m. (f) 125 to 140 m. The circles in the checkerboard test match the original dense grid used for the tomographic inversion of the real data.

The application of the W/D method to the data set provided both $V_s$ and $V_p$ models. We considered the estimated Poisson's ratio of the two clusters as prior information in the reference model of the LCI and SWT methods. Now, we use the same Poisson's ratios to transform the $V_s$ results of these two methods to $V_p$ models. We also linearly interpolate the $V_s$ and $V_p$ results from all three methods to obtain the velocity models at common voxels of $10 \times 10 \times 0.1 \text{ m}^3$ within x, y, and z (depth) directions, respectively. In Fig. 18, we compare the retrieved pseudo 3-D $V_s$ and $V_p$ models from the three methods at various iso-surfaces. Very similar trend of variations for $V_s$ (left panels) and $V_p$ (right panels) are obtained from the application of the methods, and they all depict a significant variation between the east and west side of the site.

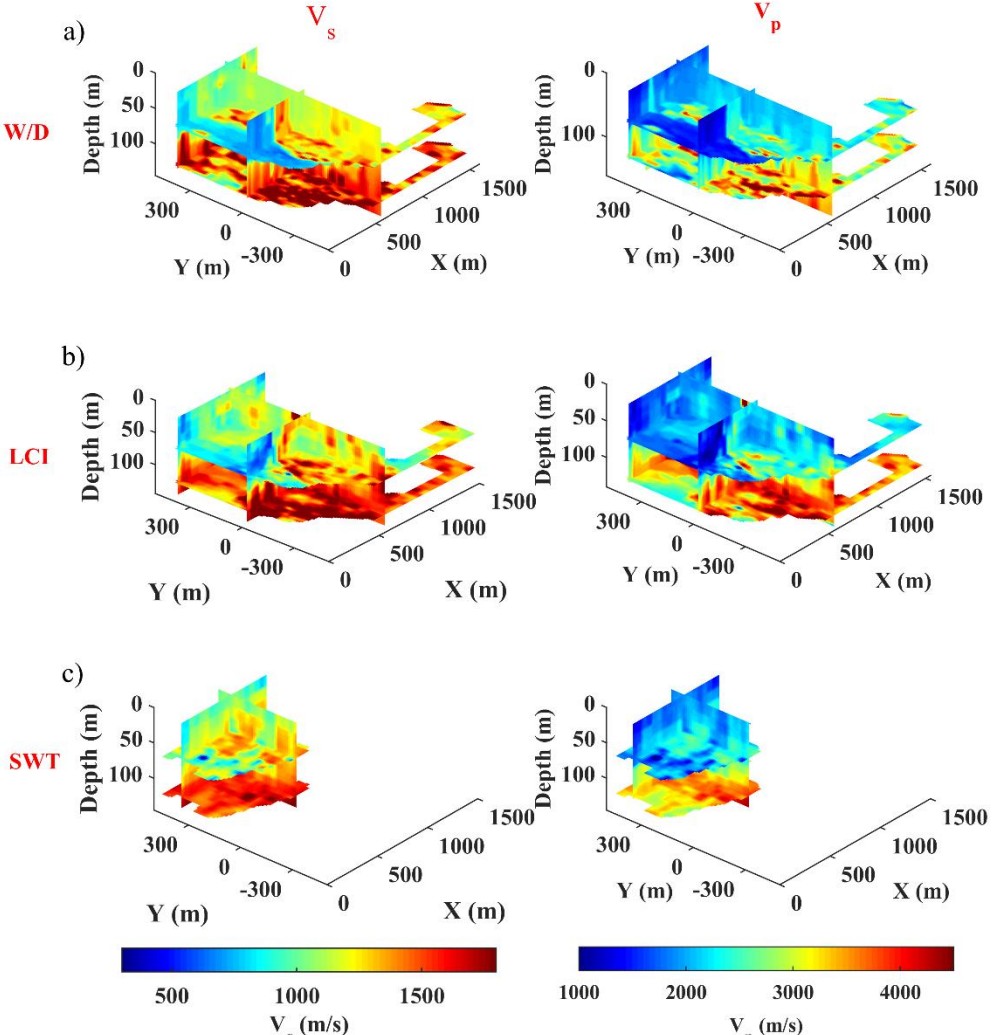

**Figure 18:** Iso-surfaces of the estimated $V_s$ models (left panel) and $V_p$ models (right panels) using: (a) W/D, (b) LCI, and (c) SWT methods. The sections are at plains x= 600 m, y =0, and 400 m, and depth = 70, and 125 m.

To compare the estimated models from each method quantitatively, we compute the difference between estimated $V_s$ and $V_p$ of every two methods separately, as:

$$\varepsilon(i,j,k) = 2 \times \left[ \frac{V(i,j,k)_{method.1} - V(i,j,k)_{method.2}}{V(i,j,k)_{method.1} + V(i,j,k)_{method.2}} \right], \tag{2}$$

where $i$, $j$, and $k$ are the indices of the voxels in x, y, and z (depth) directions, respectively, and $V$ is the velocity. In Fig. 19, we show the boxplots of the differences compartmentalized within different depth intervals. The differences are computed for depths between 20 and 140 m, except for the $V_s$ comparison of the LCI and SWT (Fig. 19e), which also

includes the first 20 m. This is due to the lack of short wavelength data for the W/D method. The significant registered differences (red "+") are mainly caused by the methods' different parameterization in depth: the W/D method provided

continuous velocities in depth (every 10 cm), while the SWT and LCI provided layered models; although we defined similar reference models for the LCI and SWT methods, the LCI was set to also update the thicknesses at each iteration, leading to different parameterization in depth compared to SWT; in addition, the W/D and LCI took as input multi-channel DCs and provided velocity models at their locations, whereas the SWT considered path-averaged DCs and resulted in velocity models in defined model point locations.

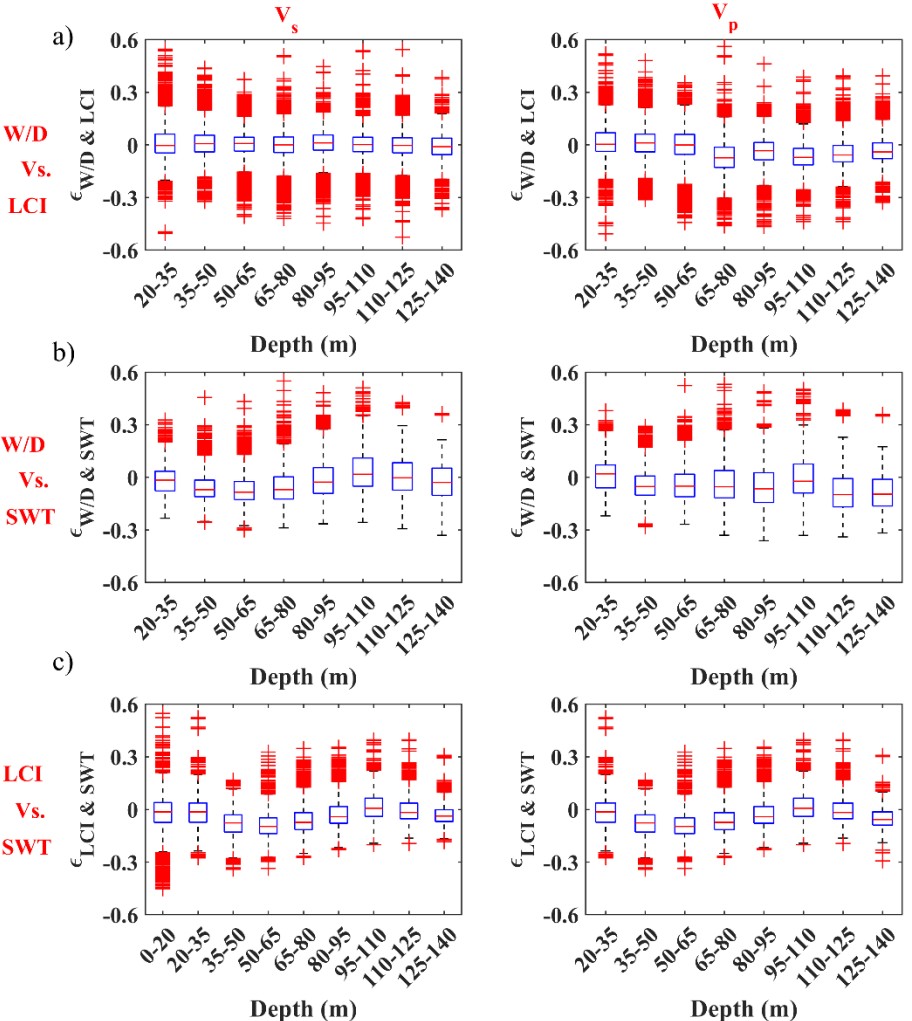

**Figure 19:** The box plot showing the difference between the $V_s$ (left panel) and $V_p$ (right panel) models obtained from the three methods computed using equation (2). The box plot is defined by three lines showing the 25th percentile, median and 75th percentile of the residual's distribution, and whisker lines extending from the box's edges up to 1.5 times the distance between the edges of the box. The rest of the data are considered as outliers and are shown with "+". The box plots show the differences between the models obtained from: (a) W/D

and LCI, (b) W/D and SWT, and (c) LCI and SWT.

The difference between the estimated $V_s$ and $V_p$ from W/D and LCI methods are small and uniform within different depth ranges (Fig. 19a). Nevertheless, for the deepest layers, over-estimation of the $V_p$ from the LCI method is registered compared to the $V_p$ from W/D technique (right panel in Fig. 19a). The differences are increased in depth when the $V_s$ and $V_p$ of the W/D and SWT methods are compared (Fig. 19b). The differences obtained for the estimated $V_s$ from the LCI and SWT methods (Fig. 19c) are very similar to the ones observed for the $V_p$, since the $V_p$ models of both methods were obtained from the estimated $V_s$ and same *a priori* Poisson's ratios; the differences are mainly less than 5%.

We compute the total differences between the estimated models of every two methods as:

$$\varepsilon_{tot} = \left[ \frac{1}{n.q} \sum_{j=1}^{q} \sum_{i=1}^{n} \left| \frac{1}{m} \sum_{k=1}^{m} \varepsilon(i,j,k) \right| \right] \times 100, \tag{3}$$

where $m$, $n$, and $q$ are the overall number of voxels in z (depth), x, and y directions, respectively. In Table 2, we report the values of the total differences obtained by comparing the models from the three methods. The total difference between the $V_s$ and $V_p$ models obtained from each pair of methods range from 3.3% to 7.06%. These differences, although not negligible, should not be regarded as substantial, considering the complexity of the near-surface. These differences are due to the presence of both significant lateral and vertical heterogeneities, as well as the different parameterization of the methods and interpolation of the models used for comparison. Given that similar a priori Poisson's ratio was used for LCI and SWT, the total difference between the two methods in terms of $V_s$ and $V_p$ is similar and the slight difference is attributed to the interpolation and parameterization. In the case of comparing the WD results with either LCI or SWT, the difference between $V_p$ models are slightly more (~1%) than $V_s$ models. The reason is that in the W/D method the $V_p$ models are obtained using the apparent Poisson's ratio. Then the interval $V_p$ models are estimated through a regularized DIX-type equation. The fact that this process is data transform creates more fluctuating results compared to layered results of the LCI and W/D.

**Table 2:** The total difference between the estimated $\boldsymbol{V_s}$ and $\boldsymbol{V_p}$ models obtained from the application of W/D, LCI, and SWT methods.

| Total difference | $\varepsilon_{tot}(V_s)$ | $\varepsilon_{tot}(V_p)$ |
|---|---|---|
| W/D vs. LCI | 3.3% | 4.67% |
| W/D vs. SWT | 6% | 7.06% |
| LCI vs. SWT | 4.74% | 4.52% |

In Table 3, we provide the approximated computational costs for each part of the three methods. The most time-consuming step of all methods is the DC estimation, which also involves expert user intervention. Compared to W/D and LCI, the SWT usually requires more DCs to reach adequate data coverage for the tomographic inversion. We estimated 1301 DCs for SWT

applied to the north of the site, whereas only 174 DCs were estimated for the application of the W/D and LCI methods to the same zone. The W/D relationship and Poisson's ratio estimation is a common stage for all three methods. The inversion running times (for LCI and SWT) given in Table 3 are for a single inversion trial using 10 CPU cores. Usually, in addition to an unconstrained inversion, several constrained inversions are performed to reach a satisfactory model in scheme of SWT and LCI methods, whereas the W/D method can be applied faster and is efficient for processing large-scale data sets. It is

noteworthy to mention that the SWT was limited to the northern zone due to computational limitations. The tomographic inversion with 1301 DCs and 300 model points was performed by a workstation equipped with 128 GB of memory and a 10-core CPU. The simultaneous inversion of both zones with at least twice the number of DCs and model points would have required exponentially higher memory and computational capabilities that our workstations could not provide. We could have under sampled the DCs and reduced the number of model points in order to invert both zones together, but we decided

to maintain the resolution of the tomographic inversion by focusing to the northern zone only.

**Table 3:** The approximated computational costs for each method.

|  | Processing (DC estimation) | W/D relationship and Poisson's ratio estimation | Model estimation |
|---|---|---|---|
| W/D | 1 min/DC | 24 hrs | 5 s/1D model |
| LCI | 1 min/DC | 24 hrs | 5 hrs |
| SWT | 1 min/DC | 24 hrs | 48 hrs |

In Fig. 20, we show the geological map superimposed with the satellite view of the area and with the horizontal slice of the estimated $V_p$ model from W/D corresponding to the depth between 35-50 m. The two diagonal and vertical faults at the north-west of the investigated area separate the east from the west. In the region between the two faults, a gap within the estimated model from the W/D method is observed: the scattering and complex propagation of surface-waves that passes through these discontinuities resulted in inconsistencies in the spectrum and prevented the estimation of reliable DCs. The

west of the area is characterized by loose formation from recent deposits (outcrop 5 in Fig. 20). The rest of the region is known for stiffer materials, composed of limestone and marl. The estimated $V_p$ also shows a higher velocity in the eastern region. The fastest $V_p$ is registered in the correspondence of the Sparnecian formation (outcrop 4 in Fig. 20). The high-velocity formation from the Danian stage that is outcropping outside of the investigated area, is probably reached below the depth of 110 m as all $V_s$ models from the three methods show very high velocity with minor lateral variations below this

depth (Fig. 11e and f, Fig. 14e and f, and Fig. 15e and f).

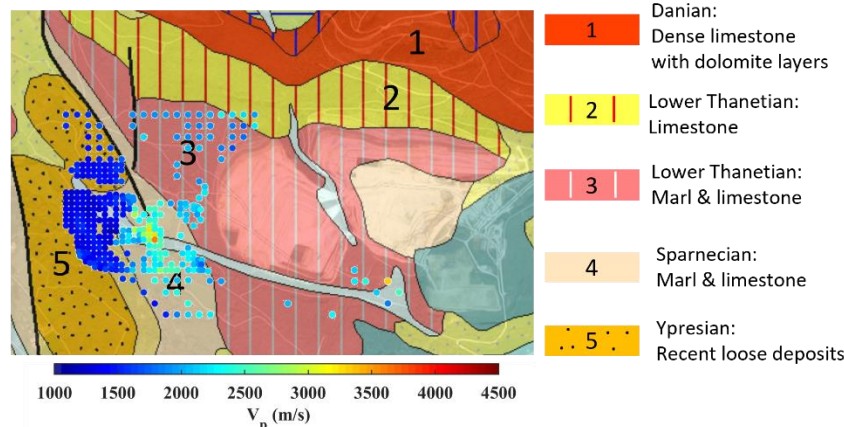

**Figure 20:** The geological map of the site, obtained from French Geological Survey (© BRGM - www.infoterre.brgm.fr), superimposed with the area's satellite view (© Google Earth) and the estimated $V_p$ corresponding to depth 35-50 m using the W/D method.

The application of the calibrated multi-channel (W/D and LCI) and two-station (SWT) methods showed promising results for the processing of the data with irregular source-receiver layout. The W/D method is cost effective and also provides $V_p$. The other two methods can provide $V_p$ only when a priori Poisson's ratio is provided. On the downside, W/D is a data transform method and the noise in the data can directly affect the results. SWT yields high-resolution models when accompanied by extensive data coverage. If the data is restricted to only a few receivers or if there is a limitation on expert resources for two-station DC picking, the performance of the SWT application can be significantly reduced. The use of LCI is highly effective in generating a laterally consistent model from a limited number of DCs. However, in the presence of significant lateral variations, the LCI is more prone to excessive smoothing. All three methods depicted a contrast between the limestone rich area in the east and loose material in the west, which are corroborated by the geological information. Given the site is an open-cast limestone mining site, the estimated models can offer valuable insights for strategizing the expansion and excavation of the quarry in the investigated zone to support the nearby cement production facility.

## 8 Conclusions

We showed the application of three surface-wave methods, W/D, LCI, and SWT, to estimate $V_s$ and $V_p$ models using a large-scale test data set obtained from a hard-rock site through irregular source-receiver recording technique. The fine-tuned multi-channel W/D and LCI methods showed potential in the processing of the low signal-to-noise ratio seismic data. Also, the irregular source-receiver outline provided high DC coverage within two-station method, facilitating high-resolution tomographic inversion. The W/D and LCI methods were applied to both zones outside the mining pits, whereas SWT application was limited to the north of the site. We used the W/D method to estimate *a priori* Poisson's ratios required for the LCI and SWT methods. The estimated $V_s$ and $V_p$ models from the three methods were less than 6% and 7.1% different, respectively. The retrieved lateral variation by the methods showed good similarity with the geological information available

for the site. The most time-consuming part of the methods are the DC pickings, especially for the SWT method which requires more DC to reach adequate data coverage compared to the other two methods. As a result, the automation of DC picking can be viewed as an important milestone in the industrialization of the surface-wave methods, facilitating their swift application to data sets even larger than the one used in this study.

*Code and data availability.* The data are licensed to Politecnico di Torino that allows research activities conducted only by Politecnico di Torino. As a result, the data cannot be made publicly available. The codes for the three surface-wave methods were developed in other projects. Nevertheless, the code for W/D data transformation method may be available by contacting the corresponding author.

*Author contributions.* FKA worked on the application of the methods to the data set with supervision of FA and LVS. FKH wrote the original paper draft, with the contribution and revisions of FA and LVS.

*Competing interests.* The contact author declares that neither him nor his co-authors have any competing interests.

*Acknowledgements*

We would like to thank Gallego Technic Geophysics for licensing the field data. The first author would like to thank TotalEnergies for supporting his PhD during which this study was carried out. We also thank the topical Editor Caroline Beghein and the anonymous reviewer for their thorough reviews and useful suggestions.

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
