# Peer review of "Comparison of surface-wave techniques to estimate S- and P-wave velocity models from active seismic data."

_EGUsphere, 2023_

## Editor Comment (EC1)

**Review of "S- and P-wave velocity model estimation from seismic surface-waves" by Anjom et al.**

This paper compares three different surface wave measurement and dispersion methods and applies them to real data. The methods include: (1) a data transformation method based on the wavelength-depth (W/D) relationship that was developed and presented by the first author in another paper; (2) a laterally constrained inversion; (3) a traditional dispersion curve analysis based on interstation measurements. I am only familiar with the third method. The advantages of the first one is that it is a lot faster than the other two and it enables to constrain VP.

Overall, the paper needs some reorganization. Too many sections and subsections that are too short. Some will need to be expanded (there is very little motivational background about the region where the methods are applied, description of the models, or geological implications). Considering the new method was already presented in a separate paper, this paper should not be yet another technique paper and needs to be more than that. It is fine to compare with other methods (though perhaps it should have been done in the previous paper), but there needs to be more than that here. It would be good to reframe the paper and focus on the geological problem that is tackled and fold the method comparison into it.

- Section 2 I a little short. It would be useful to give some more background about the location. What previous studies have been conducted? What have they found? What are the remaining issues you can solve with your technique?
- Figs. 1 and 2: Please, add latitude and longitude markers on the figures
- Section 4.2:
  - Please, specify if the Monte Carlo inversion is linear or non-linear. If it is linearized, please comment on how this can affect the solution.
  - Density is considered a priori. How? Please describe what kind of constraint is imposed. A fixed value? A fixed ratio? Something else? Justify the choice.
  - Line 145: "wide model space" is a vague statement. Please quantify by giving the bounds of your model space.
  - Have you tried to modify the bounds of the model space and see how it affects the solution?
  - $\circ$  What criteria did you apply to select the models with the Fisher test?
  - "The experimental W/D relationship is significantly sensitive to Poisson's ratio." Please, demonstrate or provide a reference to justify this statement.
  - It would be very useful to see a plot of sensitivity curves at the measured periods
  - In Figs. 6 and 7, VSZ was not defined
  - Fig. 8 is based on one model only. It would be useful to plot and discuss the uncertainties in each Vs and Vp model and how they propagate into Poisson's ratio uncertainties. Without good error estimates, differences between clusters are not meaningful
  - Section 4.3 is very short and may need to be expanded or folded into another section.
- Section 5.2
  - $\circ~~V_{P}$  and density are fixed a priori: how? Details are needed
  - Line 221: What does "contemporarily" mean in this context? I suspect it is incorrect English usage.
  - What are the lateral constraints applied?
  - Section 5.3: Please expand and describe your models
- Section 6.3: The results need to be described. A paper section should have more than one sentence.

- Section7:
  - Again, model comparison would be better with error bars on the individual models
  - Fig. 18: the caption should explain the difference between blue and red symbols
  - A discussion of the geological implications of the results is needed. The short paragraph at the end of the section needs to be expanded.
- Section 8: I think saying a method is "a great breakthrough" may be an overstatement. It would be better to use the word "advantageous" or something like that instead.

Minor comments

- Overall, a lot of indices, exponents, and subscripts need to be fixed (e.g. VS, VP, density units, etc).
- There are too many subsections that are very short and probably should be combined into bigger sections or be expanded significantly.
- Line 38: reference to Sotto et al. (2017) needs reformatting
- I am not familiar with the type of study this is applied to. Why do you calculate a time-average velocity? This needs a bit of context. Do you mean Vs measured at different times? Why would Vs depend on time?
- Line 39: Rephrase "showed with synthetic and real tests" to "showed with synthetic tests and tests on real data"
- Line 50: mantel should be mantle
- Line 53: application of SWT for the near-surface characterization  $\rightarrow$  application of SWT for near-surface characterization
- Line 54: In literature  $\rightarrow$  In the literature
- Line 60:
  - $\circ$  ~ VS should have been defined much earlier in the text
  - o "the" S-wave velocity model
  - Remove *"the recordings of"*. It is the receivers that are aligned with the event. Also, please specify that the alignment is approximately along the great-circle path and thus implies ray theory I applied.
- Line 66: "on two-station ones."  $\rightarrow$  "on two-station methods."
- Line 71: Incorrect English wording for "that is in advantage of SWT."
- Line 8: "the" south of France
- Fig. 7b: vertical axis label needs to be moved
- Line 288: analyse  $\rightarrow$  analyze

---

## Author Comment (AC1)

I am glad to review this manuscript "S- and P-wave velocity model estimation from seismic surface-waves" by Khosro Anjom et al. This work presents the application of three different surface wave methods to active-source surface wave data collected in a mining site, and estimates Vs and Vp, respectively. It's an overall good and informative paper. I have only a few comments on the details.

Dear anonymous reviewer 1,

Thank you very much for your time in reviewing our manuscript and giving useful remarks. In the following we respond to your questions and remarks.

Authors,

**Comment R1-1:** About the velocity perturbation about 7%**

To be frank, i feel like the model perturbation is a bit higher. For example, let's say fig14a, the velocity perturbation of the target structure might be smaller than 15% from my guess. Authors may calculate it and prove me wrong. If it's this case, then the uncertainty from different methods is half of local anomaly of the structure which is unacceptable. I suggest authors provide the average velocity variation at different depths and different methods, and add these information into fig 18 and table 2. It will help the reader to understand the relative scales between the uncertainty associated with methods and the real variation associated with targets.

Thank you for this comment. We should clarify that no marker formation is targeted in the estimated models. The interpolated estimated VS model between 20-35 m of depth from WD and SWT methods are 12% different (Figure 9a and Figure 14a of the submitted paper), whereas the lateral variations of the models compared to the average velocity is on average 10% and 17% for SWT and W/D models, respectively. We understand your comment regarding the model misfit and lateral variations, but we have some concerns about connecting lateral variations from the models directly to the model misfit. The model comparisons were carried under many assumptions, and they were meant to show the similarity of the models and continuity of these similarities in depth. The estimated models had completely different parameterization and the comparison was carried out after the interpolation of the estimated models. The W/D method provided the estimations every 10 cm in depth and at the location of the multi-channel DCs in X and Y directions, whereas the SWT model included 300 locations equally spaced in X and Y directions and within 7 layers of 15 m thickness (only first layer 20 m). So, some level of difference in addition to difference due to methods' accuracies is expected. In Figure R1 below, we compare the results of the VS from SWT method between 20-35 m of depth (second layer) and VS from W/D method averaged between 20-35 m. In the northern part of the area, where SWT model is superimposed to WD model, the trend of the variations is well preserved. The artifacts created by the outliers of the W/D method, which is amplified by interpolation, as well as the smooth lateral transition of the SWT model are the main contributors to the model misfits. Since the SWT model is obtained from laterally constrained inversion the transition between the high-velocity and low-velocity zones is smoother, whereas W/D model provides a sharp transition zone. Using lower lateral constraints helps in sharpening the transition zones but at the cost of creating unrealistic perturbations in the model. In the revised manuscript, we will explicitly mention the impact of the lateral constraints on the transition zone, as well as the effect of model parameterization within the scheme of the three methods.

**Figure R1:** Comparing the S-wave velocity results from surface wave tomography and W/D for depth range of 20-35 m. a) The estimated VS from SWT. B) The estimated VS from WD. C) North zone from SWT and south zone from WD.

**Comment R1-2: About the data error and model error**

I know it's challenging to collect high quality seismic data in area with stiff surface. figure 3 shows that the quality of the collected surface wave data is poor. Is it necessary to use some signal-enhance technical to denoise the data? I am afraid the data quality/error might be introduced into the final model error/uncertainty between different methods. because the SWT method uses only two-station pair which will definitely provide lower quality inputs than other two methods who employ the multi-channel inputs. I would expect some necessary discussions about this part.

Thank you for this comment. We completely agree that seismic data quality at stiff sites can be very challenging. Despite these challenges, surface wave analysis has proven to be a reliable tool for near-surface characterization in these sites. Comprehensive examples of surface wave methods application to hard rock sites can be found in Pileggi et al. (2011), hollis et al. (2018), Papadopolou et al. (2020), Da Col et al. (2020), and Colombero et al. (2022).

In test site of the submitted manuscript, thanks to the high energy vibroseis sources and to spectral stackings from different shot locations, we were able to obtain sufficient spectral resolution for DC picking. As you can see in the spectrum in Figure 4 of the submitted paper many shots are used to compute the final spectrum in Figure 4b, and the energy maxima shows continuity along the selected frequency band.

In the used carpet acquisition scheme, dense receiver grids were deployed, and sources were used only in accessible locations (along roads). This irregular source-receiver outline, as well as the exciting of the source in many locations, gave us a perfect opportunity for two-station dispersion measurements. In classical 3D measurements with regular source receiver locations, non-uniform azimuthal distribution is usually obtained, and the chance of stacking is reduced for many of the two-station configurations. Thanks to the source-receiver geometry created by carpet recording, we considered a high threshold for the number of stackings (at least five recordings from different shots for each of the two receivers). This way,

we were able to obtain good signal-to-noise ratio in the two-station analysis even though the data were from a stiff site. Nevertheless, the spectra of the multi-channel analysis provided a better signal-to-noise ratio in the low frequencies. The distribution of the data points based on wavelength (Figure 15a of the submitted paper) reflects these challenges of the two-station method in estimating low frequencies (large wavelength) data points. In the revised manuscript, we will include this information.

**Comment R1-3: About the imaging point**

I have no idea how the authors define the imaging points for the different methods. i guess the middle point will be taken as the imaging point for SWT, then what's the imaging point of the other two? please clarify this point.

Thank you for this comment. The W/D and LCI methods are based on multi-channel analysis. Within the multi-channel method, the computed dispersion curves are associated to the center of the multi-channel spread. Each DC is assumed to carry information about the local 1D properties beneath the receiver spread. In both W/D and LCI methods the models are estimated at the position of these local DCs and coincide with center of the receiver spread (blue square in figure 4a). Since the receiver spread is a moving squared window that is shifted by 1 receiver spacing in both x and y direction, the relevant DC and 1D models have the same density as the receiver grid.

On the other hand, tomographic inversion takes as input the dispersion curves retrieved from two receivers only. These curves show the path-average phase velocity along the path between the two receivers, and for each receiver couple in line with a source we can retrieve a DC. In this case, the retrieved model does not correspond to a single dispersion curve but belongs to a regular grid of 1D models which are estimated to invert all the retrieved two-station DCs at once. In Figure R2 below we have included for your convenience a map-view graphic scheme. On the left one path between two receivers superimposed to the grid of models, on the right we show that for each location, the model is estimated on the basis of many different paths. In the case of Aurignac data, 300 model points equally distant in x and y directions were used to build the model grid. Within the tomographic inversion algorithm, the synthetic path averaged DCs are computed as reciprocal of the average slowness along the paths discretized over the model grid. The phase velocities at the location of the discretized paths are computed by bi-linear interpolation of the phase velocities from adjacent model points (see the models in red in the plot on the left Figure R2). More information on how the paths is discretized and on how the synthetic DCs are computed for regularly spaced 1D models can be found in Khosro Anjom (2021). In the revised manuscript, we will describe more clearly the distinction between the model structure and parameterization obtained various methods.

For the comparison of the estimated models from the three methods in the discussion of the submitted manuscript, we interpolated the models to obtain matching geometry of the estimated models in voxels of  $10 \times 10 \times 0.1 \text{ m}^3$  in x, y and depth directions.

**Figure R2:** Schematic representation of surface wave tomography model grid definition and synthetic dispersion curve computation. a) the solid line represents the path between two stations A and B. The phase slowness of any point i along the path is determined from the values at four surrounding grid points using bilinear interpolation. b) the inversion scheme links each 1D VS profile to all experimental raypaths (solid lines) crossing its neighboring area (grey area).

**Comment R1-4: why SWT can't provide VS model at the southern zone?**

Thank you for this comment. The SWT can definitely provide a model for the southern part similar to the northern part, but we did not apply SWT to both zones due to computational capacity and memory restrictions.

Including the southern part to the surface wave tomography analysis would have meant inverting more than 3000 DCs simultaneously, given that only the north zone, which is the smaller zone, provided 1301 DCs. With more area to cover, we should have considered more model points for the inversion for a high-resolution inversion. The substantial growth in both model size and data volume would have required greater computer memory and computational capacity than what our workstations could provide. We could have under sampled the data and used a lower number of model points, but this would have impacted our study, as we wanted to perform a high-resolution inversion. As a result, we decided to focus the SWT only on the northern zone. We hope this clarifies the absence of SWT in the southern zone. In the revised manuscript, we will mention the computational constraints in performing simultaneous tomographic inversion of both zones.

Comment R1-5: 5. page 8, line 170, "Fig. 6b and c, as well as Fig. 6b and c, we show the", a typo!

Thank you. In the revised manuscript, we will fix this error.

**REFERENCES**

- Colombero, C., Papadopoulou, M., Kauti, T., Skyttä, P., Koivisto, E., Savolainen, M., and Socco, L. V.: Surface-wave tomography for mineral exploration: a successful combination of passive and active data (Siilinjärvi phosphorus mine, Finland), Solid Earth, 13, 417–429, https://doi.org/10.5194/se-13-417-2022, 2022.
- Da Col, F., Papadopoulou, M., Koivisto, E., Sito, Ł., Savolainen, M., and Socco, L. V.: Application of surface-wave tomography to mineral exploration: A case study from Siilinjärvi, Finland, Geophys. Prospect., 68, 254– 269, https://doi.org/10.1111/1365-2478.12903, 2020.
- Hollis, D., McBride, J., Good, D., Arndt, N., Brenguier, F., and Olivier, G.: Use of ambient noise surface wave tomography in mineral resource exploration and evaluation. SEG Technical Program, Anaheim, USA, Expanded Abstracts, 1937–1940, https://doi.org/10.1190/segam2018-2998476.1, 2018.
- Khosro Anjom, F.: S-wave and P-wave velocity model estimation from surface waves, PhD Thesis, Politecnico di Torino, Torino, Italy, https://iris.polito.it/handle/11583/2912984, 2021.
- Papadopoulou, M., Da Col, F., Mi, B., Bäckström, E., Marsden, P., Brodic, B., Malehmir, A., and Socco, L. V.: Surfacewave analysis for static corrections in mineral exploration: A case study from central Sweden, Geophys. Prospect., 68, 214–231, https://doi.org/10.1111/1365-2478.12895, 2020.
- Pileggi, D., Rossi, D., Lunede, E., and Albarello, D.: Seismic characterization of rigid sites in the ITACA database by ambient vibration monitoring and geological surveys. Bulletin of Earthquake Engineering 9, 1839–1854, https://doi.org/10.1007/s10518-011-9292-0, 2011.

---

## Author Comment (AC2)

This paper compares three different surface wave measurement and dispersion methods and applies them to real data. The methods include: (1) a data transformation method based on the wavelength/depth (W/D) relationship that was developed and presented by the first author in another paper; (2) a laterally constrained inversion; (3) a traditional dispersion curve analysis based on interstation measurements. I am only familiar with the third method. The advantages of the first one is that it is a lot faster than the other two and it enables to constrain VP. Overall, the paper needs some reorganization. Too many sections and subsections that are too short. Some will need to be expanded (there is very little motivational background about the region where the methods are applied, description of the models, or geological implications). Considering the new method was already presented in a separate paper, this paper should not be yet another technique paper and needs to be more than that. It is fine to compare with other methods (though perhaps it should have been done in the previous paper), but there needs to be more than that here. It would be good to reframe the paper and focus on the geological problem that is tackled and fold the method comparison into it.

Dear topical Editor,

Thank you for the time you spent reviewing our manuscript and for the useful remarks.

This site was selected to perform a research seismic campaign by Gallego Technic Geophysics and TotalEnergies to mainly test the feasibility and efficiency of the carpet recording and also to calibrate the processing tools for this type of survey layout. In a simplified explanation, carpet recording involves the use of dense receiver spread and limited number of shot locations in easily accessible places (Lys et al., 2018). It is noteworthy to mention that the carpet recording is designed for seismic exploration, and the expressions "dense receiver spread" and "limited number of shots" should be interpreted within this context. The carpet recording workflow was created to facilitate seismic measurements in remote areas such as forests and foothills. An example of carpet recording with irregular receiver grid has been also shown in (Khosro Anjom et al., 2019). The use of sources only in accessible locations makes the acquisition extremely efficient but creates irregular source-receiver layout that demands revalidation of the processing workflow. The primary aim of this study was to fine-tune, adapt, and assess the workflows for characterizing near-surface of a hard rock site when seismic data is collected through a carpet recording scheme. The possibility of retrieving reliable near-surface velocity models is important in the view of seismic processing that aim at correcting the deep exploration data for the effect of weathering layer, such as static corrections (Marsden, 1993; Cox, 1999). For this purpose, we applied three surface wave methods and compared their results with each other and with the available outcrop map. We adapted the multi-channel dispersion curve estimation to irregular source-receiver layout from the carpet recording. These dispersion curves were the inputs to WD and LCI methods. On the other hand, the irregular geometry of the source-receivers' were advantages toward the surface wave tomography and resulted in homogenous and dense azimuthal coverage, an important property for high-resolution tomographic inversion. In the revised manuscript, we will clarify in detail the motivation behind this research.

The area was chosen as a suitable place to test the carpet recording. To our knowledge, the geological information about the area is limited to the outcrop map in the manuscript that we obtained from French geological survey platform (BRGM). In the revised manuscript, we further analyze the results and draw a more comprehensive comparison to the geological map.

In the beginning, we created the outline of the manuscript so that the three methods can be easily followed by the reader. We now realize that some of the chapters are short, and it would be better to merge certain sections and add more information. These changes will be made to the revised manuscript.

In the following, we provide point-to-point response to all your remarks and comments.

Best Regards,

Authors

**Comment R2-1:** Section 2 is a little short. It would be useful to give some more background about the location. What previous studies have been conducted? What have they found? What are the remaining issues you can solve with your technique?

Thank you for your useful comment. As we mentioned above, this test site was selected to carry out the acquisition of a research experimental seismic survey mainly to test the feasibility and the efficiency of the carpet recording. To our knowledge besides the outcrop map in the submitted manuscript no other information is available and further geophysical measurements are not in reach. In the revised manuscript, we will clarify the objectives of this survey and what is the main target of the research.

**Comment R2-2:** Figs. 1 and 2: Please, add latitude and longitude markers on the figures:

Thank you for this comment. According to our agreement for the use of the data, we cannot disclose the exact latitude and altitude of the location, and of the source-receiver positions. Regarding the model estimations, we also had to scale the UTM coordinates to a new reference system for confidentiality.

**Comment R2-3:** Section 4.2:

- **Comment R2-3a:** Please, specify if the Monte Carlo inversion is linear or non-linear. If it is linearized, please comment on how this can affect the solution.

We use the 1D Monte Carlo algorithm developed by Socco and Boiero (2008). In this method, the VS, Poisson's ratio, and thicknesses are uniformly sampled within the defined model space, and the best fitting models are selected according to a statistical Fisher test. It is noteworthy to mention that the algorithm developed by Socco and Boiero (2008) takes advantage of scaling properties of dispersion curves. The synthetic DCs of the random models are computed and shifted as close as possible to the experimental DC. Then, the scaling factor is obtained from the DC shift and the models are scaled. These scaling steps, which are performed in a fully automatic manner, highly optimize the model space sampling, and reduce the number of required simulations to find the inversion results. The best fitting models are then selected according to a statistical Fisher test with a certain level of confidence. No linearization is introduced in any step of the MC inversion. Since the MC inversion is just a way to carry out one step of the workflow and the MC algorithm is widely described in the original paper, we decided not to go into the details about it.

- **Comment R2-3b:** Density is considered a priori. How? Please describe what kind of constraint is imposed. A fixed value? A fixed ratio? Something else? Justify the choice.
  It has been shown in many previous studies that even though density is a model parameter according to Haskell and Thompson's forward modelling of the surface wave dispersion curve, it has a minor impact on the simulated phase velocity (Xia et al., 1999; Foti and Strobbia, 2002; Pan et al., 2018). We assume the density according to the available information of the site. The Aurignac site is characterized by hard rock. We assumed density of 2000 kg/m$^3$ for the first layer and 2200 kg/m$^3$ for the other layers.

- **Comment R2-3c:** Line 145: "wide model space" is a vague statement. Please quantify by giving the bounds of your model space. Have you tried to modify the bounds of the model space and see how it affects the solution?

Thank you for this insightful comment. In the revised manuscript, we will provide the upper and lower boundaries of the VS model. The Poisson's ratio of the layers was sampled between 0.1 to 0.45 for all layers, and the density was fixed as mentioned in Comment R2-3b.

Yes, we have tested various boundaries not only for this data set but for other data sets in Khosro Anjom et al. (2019), Khosro Anjom (2021). The algorithm is very efficient given that it scales the sampled models according to the misfit between experimental and synthetic dispersion curves (Socco and Boiero, 2008). Hence, if the model space boundaries are inappropriate the sampling automatically gets out of them. Figure R3 below shows an example of how the models are automatically moved beyond the initial boundaries when the boundaries were selected poorly with respect to the data. Moreover, thanks to the parallel computing of the code, we sampled a large number of models within reasonable time (1,000,000 in this manuscript), which further increases the independence of the final results with respect to defined model space.

[Figure]

***Figure R3:*** *A synthetic example showing the impact of model scaling in the final results of the MCI (from Socco and Boiero, 2008). (a) and (b) 5,000 best fitting models from 200,000 sampled models for a three-layer model (magenta): (a) without scaling; (b) application of the scale properties; the homogeneous boundaries (red lines) have been on purpose selected not to contain the true model; (c) and (d) best fitting dispersion curve compared with the experimental one: (c) no scaling; (d) application of the scale properties.*

- **Comment R2-3d**: What criteria did you apply to select the models with the Fisher test?

Each synthetic dispersion curve has a misfit with respect to the experimental DC computed as (Socco and Boiero, 2008):

$$S = \frac{\sum_{i=1}^{l}\left[(v_{s,i}-v_{e,i})^2 \sigma_{e,i}^2\right]}{l-(3n-1)},$$
(R1)

where $v_{s,i}$ and $v_{e,i}$ are the elements of synthetic and experimental phase velocities, $\sigma_{e,i}$ are the elements of data uncertainty vector obtained in DC picking stage, $l$ is the number of data points, and $n$ is the number of layers in the inversion. Since the VS, Poisson's ratio and thickness are the variable of the inversion the model parameters are equal to $3n-1$. The Fisher test is applied to these misfits to select the best fitting models, considering a level of confidence. We used a low level of confidence equal to 0.001 to include in the results all geological settings that are providing a good fitting with the

experimental dispersion curves. In Figure R4 below, an example of three different levels of confidence for the inversion of an experimental dispersion curve is shown (from Socco and Boiero 2008).

[Figure]

**Figure R4:** *The selected models from the Monte Carlo inversion of an experimental dispersion curve using different levels of confidence equal to a) 0.25, b) 0.01, and c) 0.001. The figure is taken from Socco and Boiero (2008).*

- **Comment R2-3e:** "The experimental W/D relationship is significantly sensitive to Poisson's ratio." Please, demonstrate or provide a reference to justify this statement.

Thank you for this comment. Numerous tests have been performed to show the sensitivity of the W/D relationship to Poisson's ratio (Comprehensive analysis has been previously published in Socco and Comina, 2017; Khosro Anjom, 2021; Khosro Anjom et al., 2019). In Figure R5 below, as an illustration we show the synthetic simulation Khosro Anjom (2021) performed to evaluate the sensitivity of the wavelength-depth relationship to Poisson's ratio. The VS (Figure R5a) and density of the model were kept constant, and only the Poisson's ratio was changed to evaluate Poisson's ratio's impact on the wavelength-depth relationships. Figure R4b shows significant variation of the estimated wavelength-depth relationships. In this plot the color shows the Poisson's ratio corresponding to each wavelength-depth relationship

In Figure 6d and 7d of the submitted manuscript also sketches the sensitivity of the reference data for cluster 1 and 2 with respect to Poisson's ratio. In the revised manuscript, we will provide more references to this aspect.

[Figure]

**Figure R5:** *A synthetic study to evaluate the sensitivity of wavelength-depth relationship with respect to variations in Poisson's ratio (from Khosro Anjom, 2021). a) The S-wave velocity of the synthetic model.  b) the computed wavelength-depth relationships from different Poisson's ratio.*

- **Comment R2-3f:**  It would be very useful to see a plot of sensitivity curves at the measured periods

Thank you for the comment. Within the scope of the W/D data transform no inversion is carried out and the dispersion data are directly transformed to shear wave velocity models. Let us know if we misunderstood your remark or you intended this comment about the sensitivity matrices of LCI or SWT.

- **Comment R2-3g:** In Figs. 6 and 7, VSZ was not defined

Thank you for this comment. The VSZ is the time-average VS.  In the revised manuscript we replace it with time-average VS that is used in the manuscript. We also better clarify that time-average VS profiles are computed from the layered velocities.

- **Comment R2-3h:** Fig. 8 is based on one model only. It would be useful to plot and discuss the uncertainties in each Vs and Vp model and how they propagate into Poisson's ratio uncertainties. Without good error estimates, differences between clusters are not meaningful.

Thank you for this comment. Figure R6a below shows the computed Poisson's ratio cube from estimated VS and VP models of the W/D method. Separately for each cluster, we used these estimated Poisson's to obtain a standard deviation at each depth. In Figure R6b and R6c, we added these standard deviations to the estimated Poisson's ratio for each cluster as you suggested. In the revised manuscript, we will add these errorbars to the estimated Poisson's ratios of the Figure 8c and 8d of the submitted manuscript. The estimated Poisson's ratios for the two clusters are used as prior information in the LCI and SWT methods, and for the conversion of the estimated VS models from these methods to VP.

[Figure]

**Figure R6:** *Uncertainty estimation for the reference Poisson's ratios from W/D method. a) 3D view of the obtained Poisson's ratio from W/D method. b and c) Estimated Poisson's ratio from the reference estimated VS and VP model of the W/D method for cluster 1 and 2; The standard deviations were obtained from all estimated Poisson's ratio of each cluster in Figure R5a.*

**Comment R2-4:** Section 4.3 is very short and may need to be expanded or folded into another section.

We had intentionally separated the three-methods application and had kept them short to facilitate the comparison by the reader. In the revised manuscript, we merge the methods together in a single chapter. Thank you for the comment.

**Comment R2-5:** Section 5.2

- **Comment R2-5a:** VP and density are fixed a priori: how? Details are needed

As we mentioned in response to comment R2-3b, the density has minor impact on the inversion results. It is a standard approach in surface wave analysis for near-surface characterization to use fixed density based on information of the site. As we mentioned in response to comment R2-3b, we considered density of 2000 kg/m³ for the first layer and 2200 kg/m³ for the rest of the layers.

On the other hand, the VP is not fixed. As we mentioned in response to comment R2-3h, we estimated two reference Poisson's ratio for the two clusters (zones) using the W/D method (Figure R5b and R5c above). We used one of the two estimations of Poisson's ratio for each model point in scheme of LCI and tomographic inversion, based on its location. We will clarify these aspects in the revised manuscript.

- **Comment R2-5b:** Line 221: What does "contemporarily" mean in this context? I suspect it is incorrect English usage.

Maybe the word simultaneously was a more proper word as we meant that we invert the dispersion curves all together, opposed to one-by-one 1D inversions. In each iteration a single sensitivity matrix is computed and used to update all model parameters. In the revised manuscript, we will rephrase the sentence to eliminate the confusion.

- **Comment R2-5c:** What are the lateral constraints applied?

We used the same level of lateral constraints equal to 50 m/s in both LCI and SWT methods. Both algorithms are designed to allow lateral and vertical constraints, as well as constraints by a priori information. We only used lateral constraints in these inversions. Both inversions are in scheme of damped least-square deterministic scheme to minimize the misfit. The misfit $Q$ is defined as:

$$Q = \left[ (\mathbf{d_{obs}} - \mathbf{d(m)})^\mathbf{T} \, \mathbf{C_{obs}^{-1}} \, (\mathbf{d_{obs}} - \mathbf{d(m)}) \right] + \left[ (-\mathbf{R_p m})^\mathbf{T} \mathbf{C_{Rp}^{-1}} (-\mathbf{R_p m}) \right], \qquad \text{(R2)}$$

where the first term determines the misfit between the experimental data $\mathbf{d_{obs}}$ and synthetic data $\mathbf{d(m)}$. $\mathbf{m}$ is the vector of the model parameters and $\mathbf{C_{obs}^{-1}}$ is the reciprocal of the covariance matrix. The second term defines the lateral regularization of the velocities, where $\mathbf{R_p}$ is the regularization matrix composed of values 1 and -1 for the constrained parameters and zeros elsewhere. The strength of the regularization is determined by the covariance matrix $\mathbf{C_{Rp}}$.

**Comment R2-6:** Section 5.3: Please expand and describe your models

Thank you for this comment. The estimated VS model from LCI, similar to W/D estimated VS model shows a significant change in the trend of the velocity from east to west in the first 87.5 m shown in Figure 11, which is in-line with the geological information from the site (the outcrop map in Figure 19. The high velocity in the east is due to the marl and limestones formations of the Sparnecian and lower Thanetian, whereas the shallow part of the western area is covered by loose material. In the deeper portion of the model below 87.5 m the contrast almost disappears and higher velocity up to 1500 m/s is reached,

probably by reaching Danian formation that is known to have higher velocity than Sparnecian and lower Thanetian. In the submitted manuscript, we will merge the three methods sections to facilitate the description of the models, while avoiding repetitions.

**Comment R2-7:** Section 6.3: The results need to be described. A paper section should have more than one sentence.

Thank you for this comment. Similar to the estimated models from LCI and W/D methods (comment R2-6), the model from the SWT method depicted significant lateral variation between the east and west within the shallow layers. Nevertheless, the SWT model provides a smoother transition zone between east and west, which is mainly caused by the lateral constraints of the inversion. The level of constraint is a compromise between the sharpness of lateral variations and unrealistic perturbations and inconsistency in the estimated model. A weaker laterally constrained inversion will definitely enhance the sharpness of the transition zone but create unrealistic perturbations to the model (Boiero and Socco, 2010). It is noteworthy to mention that even though the constraints of the LCI and the SWT applications are similar, the SWT provides a smoother model, mainly due to the differences in the covariance matrix of the dispersion data and to the difference in the forward modelling of the synthetic dispersion curves.

Like the other two methods, the estimated VS model from SWT shows no contrast between west and east below 95 m where it reaches the Danian formation. In the revised manuscript, we will provide more description of the estimated models.

**Comment R2-8:** Section 7:

- **Comment R2-8a:** Again, model comparison would be better with error bars on the individual models.

  Thank you for this comment. For two reasons we believe the boxplots are a more reasonable choice to sketch the misfit compared to errorbars between the models from the three methods. First, they provide detailed comparison within different depth and between each two methods. Also, in each location and each depth, we have only three results that are from the three methods. As a result, obtaining an errorbar (standard deviation) for each model point at each depth may be statistically invalid. We prefer to maintain the differences between the models within the scope of the boxplots. If the topical editor believes otherwise provided these explanations, we will prepare the revisions accordingly.

- **Comment R2-8b:** Fig. 18: the caption should explain the difference between blue and red symbols.

  In the submitted manuscript, we had provided the explanation of the standard boxplot in the text of the discussion above the figures. We understand that figure caption is a better location to explain the boxplots. Basically, the box plot is defined by three lines showing the 25th percentile, median and 75th percentile of the residual's distribution, and whisker lines extending from the box's edges up to 1.5

times the distance between the edges of the box. The rest of the data are considered as outliers and are shown with "+". We moved this explanation accordingly in the revised manuscript.

- **Comment R2-8c:** A discussion of the geological implications of the results is needed. The short paragraph at the end of the section needs to be expanded.

Thank you for this comment. All three methods depict a contrast between the limestone rich area in the west and loose material in the east. These observations are corroborated by the geological information as was mentioned in comment R2-6 and R2-7. Given the site is an open-cast limestone mining site, the estimated models can offer valuable insights for strategizing the expansion and excavation of the quarry in the investigated zone to support the nearby cement production facility. In the revised manuscript, we will discuss these implications.

**Comment R2-9:** Section 8: I think saying a method is "a great breakthrough" may be an overstatement. It would be better to use the word "advantageous" or something like that instead.

Thank you for your comment. By *"As a result, the automation of the DC picking can be considered a **great breakthrough in industrialization** of these methods, which enables their fast applications to even larger data sets than the one used in this paper.",* we meant a robust automatic picking algorithm in the future can give a significant boost to the industrialization of the surface wave algorithms. To eliminate any confusion, we will rephrase the sentence as:

*"As a result, the automation of DC picking can be viewed as an important milestone in the industrialization of the surface wave methods, facilitating their swift application to data sets even larger than the one used in this study."*

**Comment R2-10: Minor comments**

- **Comment R2-10a:** Overall, a lot of indices, exponents, and subscripts need to be fixed (e.g. VS, VP, density units, etc).
  Thank you for this comment. We will fix these in the revised manuscript.
- **Comment R2-10b:** There are too many subsections that are very short and probably should be combined into bigger sections or be expanded significantly.
  We will reorganize the sections and will remove many of the subsections in the revised manuscript.
- **Comment R2-10c:** Line 38: reference to Socco et al. (2017) needs reformatting
  Thank you. We will fix this issue in the revised manuscript.
- **Comment R2-10d:** I am not familiar with the type of study this is applied to. Why do you calculate a time-average velocity? This needs a bit of context. Do you mean Vs measured at different times? Why would Vs depend on time?
  Thank you for this comment. The time-average velocity at a given depth is the average velocity of the wave (either S- or P-wave) from the surface down to that depth. The time average velocity can be used directly to obtain the traveltime of a wave from the surface down to the specific depth. The term time-average comes from the fact that to obtain an average velocity at a depth, the

traveltime within each layer should be used as the weight to the arithmetic averaging. An illustration of why it is called time-average velocity can be shown as:

$$VSZ\ (z) = \frac{\sum_n h_i}{\sum_n \frac{h_i}{VS_i}} = \frac{\sum_n VS_i.t_i}{\sum_n t_i}, \qquad (R3)$$

where, $h_i$, $VS_i$, and $t_i$ are the thickness VS and traveltime at the $i$th layer. We hope this explanation clarifies the time-average velocity concept. In the revised manuscript, we will explain more clearly the concept of time-average velocity.

The time average velocity is directly used for many purposes. For instance, in the context of seismic reflection the time average velocity is used for static corrections. These corrections are routinely applied to remove the effect of the weathering layer from the data, whose heterogeneities may represent a significant source of noise (Marsden, 1993; Cox, 1999). In the context of seismic hazard estimation, the time average velocity, also called harmonic velocity by some authors, is a proxy for local seismic response. For instance, $VS_{30}$, the time average velocity at 30 m depth, is a standard parameter, used in many national and international regulations, to classify sites according to seismic response classes, thanks to the correlation between $VS_{30}$ and PGA (peak goring acceleration). In the W/D method, to retrieve the W/D relationship, the layered model is transformed into its corresponding time-average model (equation R3) and compared with the DC plotted as a function of wavelength.

- **Comment R2-10e:** Line 39: Rephrase "showed with synthetic and real tests" to "showed with synthetic tests and tests on real data"
  Thank you for this comment. We will implement the change in the revised manuscript.

- **Comment R2-10f:** Line 50: mantel should be mantle
  Thank you for this comment. We will implement the change in the revised manuscript.

- **Comment R2-10g:** Line 53: application of SWT for the near-surface characterization➔ application of SWT for near-surface characterization
  Thank you for this comment. We will implement the change in the revised manuscript.

- **Comment R2-10h:** Line 54: In literature ➔ In the literature
  Thank you for this comment. We will implement the change in the revised manuscript.

- **Comment R2-10i:** Line 60:
  - VS should have been defined much earlier in the text
  - "the" S-wave velocity model
  - Remove "the recordings of". It is the receivers that are aligned with the event. Also, please specify that the alignment is approximately along the great-circle path and thus implies ray theory I applied.
  Thank you for this comment. We will implement the change in the revised manuscript.

- **Comment R2-10j:** Line 66: "on two-station ones." à "on two-station methods."
  Thank you for this comment. We will implement the change in the revised manuscript.

- **Comment R2-10k:** Line 71: Incorrect English wording for "that is in advantage of SWT."

  Thank you for this comment. We will implement the change in the revised manuscript.
- **Comment R2-10l:** Line 8: "the" south of France

  Thank you for this comment. We will implement the change in the revised manuscript.

- **Comment R2-10m:** Fig. 7b: vertical axis label needs to be moved

  Thank you for this comment. We will implement the change in the revised manuscript.

- **Comment R2-10n:** Line 288: analyse ➔ analyze

  Thank you for this comment. We will implement the change in the revised manuscript.

**REFERENCES**

Boiero, D., and Socco, L.V.: Retrieving lateral variations from surface wave dispersion curves, Geophysical Prospecting, 58, 977-996, https://doi.org/10.1111/j.1365-2478.2010.00877.x, 2010.

Cox, M.: Static Corrections for Seismic Reflection Surveys, SEG, 546, 1999.

Foti, S., and Strobbia, C.: Some notes on model parameters for surface wave data inversion: Symposium on the Application of Geophysics to Engineering and Environmental Problems (SAGEEP), https://doi.org/10.4133/1.2927179, 2002.

Khosro Anjom, F.: S-wave and P-wave velocity model estimation from surface waves, PhD Thesis, Politecnico di Torino, Torino, Italy, https://iris.polito.it/handle/11583/2912984, 2021.

Khosro Anjom, F., Teodor, D., Comina, C., Brossier, R., Virieux, J., and Socco, L. V.: Full-waveform matching of VP and VS models from surface waves, Geophys. J. Int., 218, 1873–1891, https://doi.org/10.1093/gji/ggz279, 2019.

Lys, P.O., Elder, K., Archer, J., and the METIS Team: METIS, a disruptive R&D project to revolutionize land seismic acquisition, in: RDPETRO 2018: Research and Development Petroleum Conference and Exhibition, Abu Dhabi, UAE, 9-10 May 2018, 2018.

Marsden, D.: Static corrections--a review, Part 1: The Leading Edge, 12, 43-49, doi: 10.1190/1.1436912, 1993

Pan, L., Chen, X., Wang, J., Yang, Z., Zhang, D.: Sensitivity analysis of dispersion curves of Rayleigh waves with fundamental and higher modes, Geophysical Journal International, 216, 1276–1303, https://doi.org/10.1093/gji/ggy479, 2019.

Socco, L. V. and Comina, C.: Time-average velocity estimation through surface-wave analysis: Part 2 — P-wave velocity, Geophysics, 82, U61–U73, https://doi.org/10.1190/geo2016-0368.1, 2017.

Socco, L.V., and Boiero, D.: Improved Monte Carlo inversion of surface wave data: Geophysical Journal International, 56, 357-371, doi: 10.1111/j.1365-2478.2007.00678.x, 2008.

Xia, J., Miller, R.D., and Park, C.B., Estimation of near-surface shear-wave velocity by inversion of Rayleigh waves, Geophysics, 64, 691-700, https://doi.org/10.1190/1.1444578, 1999.

---

## Author Response (AR2)

Reviewer 1:

**This paper applies three different methods to real seismic data to estimate both Vs and Vp near-surface structures. The wavelength-depth method gives the Vp structure, while all three methods give Vs structures. As a reviewer joined from the revised version, I feel that the qualities of the sentences and figures are sufficient for the publication. Below are some comments that should be addressed before the publication.**

Dear anonymous reviewer 1,

We appreciate the time you spent reviewing our manuscript and we think that your remarks were very useful to clarify the scope and the outcomes of our work. Below, we respond point by point to your questions and remarks.

Sincerely,

Authors

**Comment R1-1: The title is too broad or general. This title can be used for a variety of studies. There should be more specific words to identify this work.**

Thank you for this comment. We change the title to "**Comparison of surface wave techniques to estimate S- and P-wave velocity models from active seismic data**" to provide a clearer explanation of the objectives of the manuscript.

**Comment R1-2: The method for estimating Vp should be described more. I guess everyone needs to see Socco and Comina (2017). It is helpful if the data, model parameters (number of layers; yes, I realized that there is no "layers" for the W/D method later, but that was really confusing before realizing it), and outputs are clearly stated in each method.**

Thank you for this comment. This study examines the implementation of three methodologies that have been previously documented by different authors. The primary aim of the paper is to demonstrate the capabilities of non-standard methods in seismic surveys for reconstructing the near-surface using data that has an irregular source-receiver layout. The readers are referred to original works for better understanding of the methods. However, as you suggested, we have included additional sentences in the explanation of the W/D method to enhance the flow and understanding of the paper.

**Comment R1-3: The disadvantages of W/D method should be described in the introduction section. In my understanding, up to ~10% uncertainty exists for both Vs and Vp estimations.**

**This uncertainty is significant compared to conventional inversion methods. The W/D method seems like an approximation (transformation) method.**

Thank you for this comment. Table 2 in the submitted paper demonstrates that the average difference between the models obtained by the different methods ranges from 3.3% to 7% for S-wave and P-wave velocity models. The near-surface environment is typically characterized by significant heterogeneity in both lateral and vertical directions. Moreover, the inversion of surface-wave data exhibits non-uniqueness, thereby contributing to the differences in the output of the methods. Furthermore, different parameterizations were employed in each of the three models. The W/D, being based on a data transform, provides velocity models with vertical discretization of 1 m intervals, whereas the SWT and LCI give layered models based on the parameterization defined for the inversion. W/D and LCI provided models at the location of the multichannel dispersion curves, while SWT provided the velocity models at the predefined locations of model points. The layer thickness of the models were defined a priori and were fixed in SWT, while LCI algorithm updated also the thicknesses. The models were compared after interpolating them to corresponding spatial voxels. Considering these factors (i.e., heterogeneity of the near-surface, inversion non-uniqueness, different parameterization of the methods and interpolation of the 3D models), average misfit of 3.3% to 7% among the VS and VP from the various methods should not be considered high. We hope this explanation clarifies the extent of the similarities. In the revised manuscript, we have reported this explanation to further clarify the factors that contribute to the differences in the models.

**Comment R1-4: The main purpose of this study was not clear and should be clarified in both the abstract and introduction. Why did you compare the three methods? Is it to prove that the W/D method gives Vs structure comparable to conventional methods despite the theoretical uncertainties (and also gives Vp structure in less computational cost)?**

Thank you for this question. The objective of the paper are three folds:

1- SWT is a very well-known technique in seismology, but it is rarely used for near-surface high-resolution modelling. The W/D method is a new method that provides the P-wave velocity in addition to S-wave velocity and requires very limited inversion and computational efforts. The LCI is a powerful tool that has not yet been truly exploited for its potential in practical applications. The comparison between the performance of the three methods on a 3D velocity model has never been done to our knowledge and provides an insight in the potential of surface-wave in the estimation of near-surface models.

2- The surface wave methods were rarely used in hard rock sites due to the low signal-to-noise ratio. But we showed that with proper workflow, the three surface wave methods can successfully be applied for reconstruction of the near-surface.

3- Finally, most surface wave methods have been customized for regular 2D acquisition setup where source and receiver locations are inline. In this study, we show that the multi-channel methods (i.e., LCI and W/D) can successfully be applied to data with irregular source-receiver locations which are not common in near surface applications. Also, this irregularity is to the advantage of the SWT data that usually suffers from low and inhomogeneous data coverage originated by in-line source location footprint.

We modified the abstract and introduction of the manuscript to better highlight these motivations for the manuscript.

**Comment R1-5: The spatial distribution of Poisson's ratio should be shown in addition to the Vs and Vp structures. It is not clear whether the estimated ratio gives meaningful a priori information to the conventional surface-wave analyses.**

Thank you for this comment. Prior research has demonstrated the efficiency of clustering in identifying lateral variations (**Khosro Anjom et al., 2017; Khosro Anjom 2021**), as well as the ability to accurately predict the VP models from VS using Poisson's ratio of the cluster (**Socco and Comina, 2017; Khosro Anjom et al., 2019**). Nevertheless, in the revised manuscript, we include a subplot (Figure 13e) of the 3D Poisson's ratio estimation from W/D method. The uncertainties for the reference layered Poisson's ratio in Figures 13c and 13d were computed from the 3D Poisson's ratio in Figure 13e.

**Comment R1-6: Why are the differences in three Vp models larger than that of Vs even though a priori information is common?**

Thank you for highlighting this aspect. The a priori Poisson's ratios are utilized in LCI and SWT. Consequently, a nearly identical difference is observed when comparing the VS and VP models from LCI and SWT techniques (4.52% and 4.74%; Table 2 of the manuscript). The slight difference is due to the interpolation of the 3D models into similar spatial voxels and different parameterization during inversion.

In the case of comparison of WD results with either LCI or SWT, the difference between VP models is slightly more (~1%) than VS models. The reason is that in the W/D method the VP models are obtained using the apparent Poisson's ratio. The apparent Poisson's ratios retrieved from the W/D sensitivity analysis are applied to estimate the time-average VP and then the interval VPs are estimated through a regularized DIX-type equation. The fact that this process is data transform creates more fluctuating results compared to layered results of the LCI and W/D. These factors and also the interpolation of the models lead to a slightly higher difference (about 1%) between the VP comparison of the W/D with respect to LCI and SWT. In the discussion of the revised manuscript, we highlight clearly this aspect.

**Comment R1-7: Some words were not familiar to me: carpet, "interval" models, and "time-average" models. Maybe because I'm not a native speaker and/or the research field is slightly different.**

Carpet recording is a new paradigm of data collection introduced by TotalEnergies. Basically, in this acquisition scheme, in contrast to classical 3D seismic exploration acquisitions, the receivers are placed over a regular grid, and the sources are used only in accessible locations. This approach creates an irregular source-receiver layout that needs to be tested for the surface wave methods applications. In fact, one of the objectives of this study is to show the application of the three surface wave methods to the data acquired in this manner. Since the second reviewer also asked for more details about the "carpet recording" terminology, and for more simplicity, we remove the term

"carpet recording" in the revised manuscript. Instead, we use irregular receiver-source acquisition layout.

The concept of interval velocity is a standard concept in seismic reflection processing. According to the definition given in Applied Geophysics Dictionary by Sheriff (2002) the interval velocity is described as: "The velocity of an interval in the subsurface measured by determining the traveltime over a depth interval along some raypath. The interval velocity is often used for velocity calculated by the Dix Formula". In a locally 1D velocity model the interval velocity is the velocity of a layer between two depth levels within which the velocity is considered constant. In our manuscript we use the term interval velocity to identify the velocity of the 1m intervals used to discretize the W/D models. We instead use the term layered velocity for the velocity of the layers of layered models used in the inversion of SWT and LCI inversion.

The time-averaged velocity at a specific depth refers to the average velocity of either an S-wave or a P-wave from the surface to that particular depth and can be directly utilized to calculate the time required to travel from the surface to that depth along a vertical path. The term "time-average" refers to the fact that the travel time within each layer is used as a weight for arithmetic averaging of the velocity of all the layer down to the depth of interest. An exemplification of the rationale behind the term "time-average velocity" can be demonstrated as follows:

$$VSZ\,(z) = \frac{\sum_n h_i}{\sum_n \frac{h_i}{VS_i}} = \frac{\sum_n VS_i.t_i}{\sum_n t_i}, \qquad\qquad (R1)$$

where, $h_i$, $VS_i$, and $t_i$ are the thickness VS and traveltime at the $i$th layer. We hope this explanation clarifies the time-average velocity concept.

The time-averaged velocity is extensively utilized for various purposes. In seismic reflection, the time-average velocity is employed for computing static corrections. The data is regularly corrected to eliminate the impact of the weathering layer, which can introduce significant noise due to its heterogeneities (Marsden, 1993; Cox, 1999). Within the framework of seismic hazard estimation, the time-average velocity, also known as harmonic velocity in certain literature, serves as a proxy for peak ground acceleration. For example, VS30, which represents the time-average velocity at a depth of 30 meters, is a widely used standard parameter in national and international regulations.

In exploration seismic these terms are common. In order to target a wider range of readers, we included additional explanatory phrases in the introduction to enhance comprehension and differentiation of the layered, interval, and time-average velocities.

**Reviewer 2**

**This paper compares and contrasts three different surface-wave methodologies applied to the same study area and dataset. The description of each method is extensive and most of the required detail is included in the manuscript, however, the link to the geological implications for the study area itself appears to be lacking. What has each result revealed about the study area? Based on the analysis is there a preferred technique for tackling active-source surface-wave imaging in studies such as these? Overall, the manuscript is well-written and informative. Below I detail some mostly minor comments.**

Dear Anonymous reviewer 2,

We appreciate the time you dedicated to evaluating our manuscript and we think that your remarks helped improve the manuscript. As previously stated in the initial round of revision, the geological data for the site is confined to the outcrop map presented in the paper. The manuscript aims to demonstrate the efficiency of utilizing three-surface wave methods for analyzing data obtained through non-standard acquisition layouts. None of the proposed methods, even though already presented in publications, are standard in near-surface applications and their comparison on a fairly large dataset and a 3D velocity model was never carried out to our knowledge. In active surface wave analysis, moreover, the P-wave velocity model, which is typically not retrieved, can be obtained using the W/D method due to its sensitivity to Poisson's ratio.

A comprehensive comparison of the methods is provided in the discussion. We prefer not to choose a specific method, but instead, we highlight the benefits and drawbacks that can aid in the selection of a method based on the site, acquisition, and resources. The W/D method is cost effective and also provides VP. The other two methods can provide VP only when a priori Poisson's ratio is provided. On the downside, W/D is a data transform method and the noise in the data directly affect the results. SWT yields high-resolution models when accompanied by extensive data coverage. If the data is restricted to only a few receivers or if there is a limitation on expert resources for two-station picking, the performance of the SWT application can be significantly reduced. The use of LCI is highly effective in generating a laterally consistent model from limited number of dispersion curves. However, when dealing with media with significant lateral heterogeneities, the LCI is more susceptible to excessive smoothing. In the discussion of the revised manuscript, we included these aspects about the performance of the three methods.

In the following, we respond point-by-point to your comments/questions in the following.

Regards,

Authors

**Comment R2-1: In the introduction it is not clear if group or phase velocities are used in the surface-wave tomography, though this is made clear further down in the method sections. I would suggest clarifying this in the introduction too, perhaps on Page 2, Line 30.**

We appreciate this comment. We clarified in page two of the revised manuscript that we use phase velocity of surface waves.

**Comment R2-2: Is it possible to plot sensitivity kernels for surface-waves as a function of depth? E.g., phase velocity sensitivity as a function of period/frequency and depth? This would help indicate what depths the dispersion curves are sensitive to. Have a look at Figure 4c in Darbyshire et al., (2013; EPSL).**

Thank you for this comment. We believe the checkerboard test shows the sufficient sensitivity of the data to the investigation depth. Since we prefer to keep the flow of the manuscript, we provide the sensitivity plot you requested here (Figure R1). We performed sensitivity kernel for a random 1D model at last iteration of the SWT, for different periods (frequencies) between 0.25 s to 0.04 s (4 Hz to ~30 Hz). Similarly in Figure R2, we show an image of the sensitivity Kernel between 4 Hz to 30 Hz normalized at each frequency. Confirming the results of the checkerboard test, the sensitivity kernel shows good sensitivity both for shallow and deeper portions. As was also depicted by the checkerboard test, the sensitivity decreases in deeper portions. We prefer not adding these figures to the manuscript because we do not think they will add information with respect to the checkerboard test and they would make the paper lengthy. We leave to the topic editor the decision if these figures are needed.

[Figure]

***Figure R1:*** Surface wave sensitivity Kernel for various periods corresponding to frequencies between 4 Hz to 30 Hz.

[Figure]

**Figure R2:** Surface wave sensitivity Kernel for frequencies between 4 Hz to 30 Hz, matching the frequency range of the data. The sensitivities are normalized at each frequency

**Comment R2-3: My background is in passive seismology so I'm not quite clear on what a carpet recording method. A sentence or two to explain what this is in more depth would be useful. Perhaps on Page 2, Line 44.**

Carpet recording acquisition technique was created by TotalEnergies to enable active data acquisition in remote areas such as foothills or densely vegetated forests. Basically, in this technique, in contrast to classical 3D active data acquisition, the receivers are deployed over dense regular or irregular grid and the sources are limited to accessible locations (Lys et al., 2018). In the case of Aurignac data, the grid of receivers was regular with variable spacing. However, due to the fact that the sources were only used along the roadsides, an irregular layout of receivers and sources was obtained. In fact, one of the objectives of the study is to adapt and test surface-wave methods in situations where the layout of source and receivers are irregular. Since reviewer one also had remarks about the term "carpet recording", we removed it entirely from the manuscript and replaced it with irregular receiver-source layout in the manuscript.

**Comment R2-4: I was going to suggest adding latitude/longitude values to Figure 1 but noticed the authors response to reviewer 2's similar comment. Nevertheless, is there a way to better link Figures b and d? Do they correspond to exactly the same area? Perhaps add a box outline in figure 1b to show what region figure 1d corresponds to.**

Thank you for the comment. As previously stated, it is not possible to include the coordinates in the figures. However, as you suggested, we included a box that shows the boundaries of the acquisition area over the outcrop map (Figure 1c of the revised manuscript).

**Comment R2-5: Page 4, Line 104: "918 receivers". Might be useful to include some more instrumentation detail here. What kind of receivers?**

The information about the receivers including the type of receiver, sampling rate and time window are summarized in Table 1. Five Hz vertical geophones were used, and the data was sampled every 2 ms for a duration of 5 seconds. In addition, 24-ton vibrator source was excited in 1077 locations. Each sweep lasted for 24 s, between 3 to 110 Hz, with 5 s of listening time. All this information is available in the site description and field data set section (above Table 1) of the manuscript.

**Comment R2-6: Page 5, Line 126: "The data exhibits a low signal-to-noise ratio as expected for hard rock sites." I would have thought soft rock sits have a low signal-to-noise ratio, whereas hard rocks allow for high signal-to-noise-ratio because of less attenuation? Perhaps add a citation for this statement since you mention this is 'expected'.**

Rayleigh waves tend to develop more in media with regular vertical velocity gradient than in stiff materials characterized by local heterogeneities that can create scattering and exhibit poor dispersion. Hence, in soft soils, dispersion curves are usually smooth and broad band with clear mode separation. In rock sites they tend to be noisy and narrow banded. The two examples of f-k spectra and relevant dispersion curves in Figures R3 and R4 were produced from a seismic line in a desert area where the acquisition line crosses both sand dunes and fractured rock outcrops. The acquisition was carried out with a heavy vibroseis. The data from the loose sand zone provide very high-quality dispersion curves while the data from the outcrop zone generate very noisy dispersion curves. Similar results have been obtained in several cases of acquisition on hard rock sites where the dispersion curves are poorly dispersive and very noisy. This aspect has been extensively addressed for instance in the PhD thesis of Papadopoulou (2021). We added this reference to the revised manuscript to support our claim.

[Figure]

**Figure R3:** Example recorded surface-waves from soft loose sand (sand dunes): (left) f-k spectrum. (right) multimodal dispersion curves

[Figure]

***Figure R4:*** Example recorded surface-waves from the same site of Figure R3 (same acquisition line) but on an outcrop: (left) f-k spectrum. (right) multimodal dispersion curves

**Comment R2-7: Page 8, Lines 179-180 and Figure 6b:**

**It still looks like there is almost double the paths coming from 0-40 degrees and 140-180 degrees than all the other azimuths. Perhaps run a surface-wave tomographic inversion excluding the excess paths in those directions and see if the result is similar. I.e., remove a subset of paths from the most sampled directions and see how that changes the final surface-wave tomography models.**

Thank you for this comment. We believe the level of inhomogeneity in the Aurignac dispersion curve data coverage does not have any significant effect on the inversion process, as indicated by the absence of any directional pattern in the checkerboard (Figure 17 of the revised manuscript). We have previously analyzed data that exhibited a strongly inhomogeneous azimuthal illumination. In these instances, both the checkerboard test and the models demonstrate directionality. Figure R5 displays an illustrative example extracted from a previously published study by Khosro Anjom et al. (2021). The limited data coverage between 20 to 140 degrees (Figure R5a) resulted in reduced weights within these directions in the inversion process and led to the creation of a directional model (Figure R5b). Similar directionality can be observed in the checkerboard test on the same data (Figure R6). The tomographic inversion was sensitive to the dominant direction, while model showed poor sensitivity to directions with low data coverage.

The estimated model and checkerboard test (Figure 15 and 17 of the revised manuscript) for the Aurignac data show no indication of directionality caused by nonuniformity of the data coverage. Therefore, in this scenario, we believe the act of decreasing the data for the purpose of achieving more uniform data coverage would not yield any advantages and is likely to diminish the level of detail in the model. Due to the limited time available for revision, we are unable to carry out another

inversion to demonstrate this aspect. However, if the topical editor believes this test is crucial, we can carry it out the requested inversion within an appropriate revision timeframe.

[Figure]

**Figure R5:** The impact of significantly nonuniform azimuthal illumination on the tomographic inversion (reproduced from Khosro Anjom et al., 2021): (a) Azimuthal illumination. (b) Estimated VS model between 20 m to 30 m of depth.

[Figure]

**Figure R6:** The checkerboard test performed on the data from figure R5 (reproduced from Khosro Anjom et al., 2021): (a) The perturbation pattern. (b) The recovered perturbation after inversion.

**Comment R2-8: Page 11, Lines 249-251: Is there a figure for this? Like a trade-off curve showing data misfit versus level of constraint, for example, and the ideal model chosen along the curve?**

The approach involves performing multiple constrained and unconstrained inversions in order to obtain a constraint level that provides consistent model, while achieves a data misfit not significantly higher than unconstrained inversion. This approach (Boiero and Socco, 2011) serves as a crucial measure for reducing the excessive smoothing in models. In the case of LCI inversion, the unconstrained inversion yielded an L2 weighted misfit of 23.4, whereas the selected constrained inversion resulted in a misfit of 23.9. This indicates that the constrained inversion does not

excessively smooth the model. Similarly, the unconstrained and selected constrained tomographic inversions yielded 42.1 and 43 L2 misfits, respectively. It is noteworthy to mention that the misfits are computed based on equation R1 (please see response to comment R2-9).

We do not employ the technique of cross-cutting between unconstrained and constrained inversion misfits to determine the level of constraint. Instead, we compare the misfit values of the desired constrained model with the unconstrained model to check if it is not excessively smoothened. Therefore, we prefer not to include an additional plot of misfit. Nevertheless, in the updated manuscript, we report the misfit values obtained from both unconstrained and selected constrained inversions.

**Comment R2-9: Was regularization used such as damping and smoothing in the inversions? It isn't immediately clear in the method section if this was done.**

Thank you for this comment. We incorporate lateral constraints within damp least square inversion scheme (Marquart,1963). According to this inversion scheme that is the same for LCI and SWT, the misfit is computed as:

$$Q = \left[ (\mathbf{d_{obs}} - \mathbf{d(m)})^{\mathbf{T}} \, \mathbf{C_{obs}^{-1}} \, (\mathbf{d_{obs}} - \mathbf{d(m)}) \right] + \left[ (-\mathbf{R_p m})^{\mathbf{T}} \mathbf{C_{Rp}^{-1}} (-\mathbf{R_p m}) \right], \tag{R1}$$

where the first term determines the misfit between the experimental data $\mathbf{d_{obs}}$ and synthetic data $\mathbf{d(m)}$. $\mathbf{m}$ is the vector of the model parameters and $\mathbf{C_{obs}^{-1}}$ is the reciprocal of the covariance matrix. The second term defines the lateral regularization of the velocities and thicknesses, where $\mathbf{R_p}$ is the regularization matrix composed of values 1 and -1 for the constrained parameters and zeros elsewhere. The strength of the regularization is determined by the covariance matrix $\mathbf{C_{Rp}}$.

$$\begin{aligned} \mathbf{m_{n+1}} = \ & \mathbf{m_n} + \left( \left[ \mathbf{G^T C_{obs}^{-1} G} + \mathbf{R_p^T C_{Rp}^{-1} R_p} + \lambda \mathbf{I} \right]^{-1} \right. \\ & \left. \times \left[ \mathbf{G^T C_{obs}^{-1} (p_{obs} - p(m))} + \mathbf{R_p^T C_{Rp}^{-1} (-R_p m_n)} \right] \right), \end{aligned} \tag{R2}$$

where $\mathbf{G}$ the Jacobian matrix, evaluates the sensitivity of the dispersion curves to the model parameters. $\mathbf{m_n}$ and $\mathbf{m_{n+1}}$ are the previous and updated model vectors, respectively.

We hope this comment clarifies the inversion method. We rather no to include this information in the revised manuscript as the method is published. The details for the LCI inversion can be found in Socco et al. (2009), while Boiero (2009) and Khosro Anjom (2021) provide comprehensive explanation regarding the tomographic inversion formulation. Nevertheless, for both LCI and SWT we clearly mention in a sentence the used inversion approach.

**Comment R2-10: Page 17, Line 348: "dense model grid". More description of the model grid is needed. E.g., what is the spacing of the nodes etc. Perhaps including Figure R2 from the response to reviewers in the main text would be beneficial.**

Thank you for this comment. The grid density can be observed in the plots of the checkerboard test (Figure 17). To preserve the coherence and continuity of the manuscript, we prefer not to introduce an additional subplot in Figure 15 of the revised manuscript, so as to avoid any unnecessary complexity. Instead, we added a clear sentence in both the caption and explanation of Figure 17 that

the circles of the checkerboard test match exactly the dense model grid used for the SWT in Figure 15 of the revised manuscript.

**REFERENCES**

Boiero, D.: Surface wave analysis for building shear wave velocity models, Ph.D. thesis, Politecnico di Torino, 233 pp., 2009.

Boiero, D., Socco, L.V.: Retrieving lateral variations from surface wave dispersion curves, Geophys. Prosp., 58, 977-996, https://doi.org/10.1111/j.1365-2478.2010.00877.x, 2010.

Cox, M.: Static Corrections for Seismic Reflection Surveys, SEG, 546, 1999.

Khosro Anjom, F.: S-wave and P-wave velocity model estimation from surface waves, PhD Thesis, Politecnico di Torino, Torino, Italy, https://iris.polito.it/handle/11583/2912984, 2021.

Khosro Anjom, F., Teodor, D., Comina, C., Brossier, R., Virieux, J., and Socco, L. V.: Full-waveform matching of VP and VS models from surface waves, Geophys. J. Int., 218, 1873–1891, https://doi.org/10.1093/gji/ggz279, 2019.

Khosro Anjom, F., Browaeys, T. J., and Socco, L. V.: Multimodal surface-wave tomography to obtain S- and P-wave velocities applied to the recordings of unmanned aerial vehicle deployed sensors, Geophysics, 86, R399–R412, https://doi.org/10.1190/geo2020-0703.1, 2021.

Khosro Anjom, F., Arabi, A., Socco, L.V. and Comina, C.: Application of a method to determine S and P wave velocities from surface waves data analysis in presence of sharp lateral variations, 36th GNGTS national convention, 632-635, https://hdl.handle.net/11583/2740539, 2017.

Lys, P.-O., Elder, K., Archer, J., and the METIS Team: METIS, a disruptive R&D project to revolutionize land seismic acquisition, in: RDPETRO 2018: Research and Development Petroleum Conference and Exhibition, Abu Dhabi, UAE, 2018.

Marquart, D.: An algorithm for least squares estimation of nonlinear parameters, Journal of the Society of Industrial Applied Mathematics, 2,431-44, https://doi.org/10.1137/0111030,1963.

Marsden, D.: Static corrections--a review, Part 1, The Leading Edge, 12, 43-49, doi: 10.1190/1.1436912, 1993

Papadopoulou, M.: Surface-wave methods for mineral exploration, Ph.D. thesis, Politecnico di Torino, 2021.

Sheriff, R.: Encyclopedic Dictionary of Applied Geophysics, SEG, http://dx.doi.org/10.1190/1.9781560802969

Socco, L. V. and Comina, C.: Time-average velocity estimation through surface-wave analysis: Part 2 — P-wave velocity, Geophysics, 82, U61–U73, https://doi.org/10.1190/geo2016-0368.1, 2017.

Socco, L. V., Comina, C., and Khosro Anjom, F.: Time-average velocity estimation through surface-wave analysis: Part 1 — 585 S-wave velocity, Geophysics, 82, U49–U59, https://doi.org/10.1190/geo2016-0367.1, 2017.

Socco, L. V., Boiero, D., Foti, S., and Wisén, R.: Laterally constrained inversion of ground roll from seismic reflection records, Geophysics, 74, G35–G45, https://doi.org/10.1190/1.3223636, 2009.